# Spherical Sliced-Wasserstein

**Clément Bonet[1], Paul Berg[2], Nicolas Courty[2], François Septier[1], Lucas Drumetz[3], Minh-Tan Pham[2]**
Université Bretagne Sud, LMBA[1], IRISA[2]; IMT Atlantique, Lab-STICC[3]
`{clement.bonet, paul.berg, francois.septier}@univ-ubs.fr`
`{nicolas.courty, minh-tan.pham}@irisa.fr;lucas.drumetz@imt-atlantique.fr`

## Abstract

Many variants of the Wasserstein distance have been introduced to reduce its original computational burden. In particular the Sliced-Wasserstein distance (SW), which leverages one-dimensional projections for which a closed-form solution of the Wasserstein distance is available, has received a lot of interest. Yet, it is restricted to data living in Euclidean spaces, while the Wasserstein distance has been studied and used recently on manifolds. We focus more specifically on the sphere, for which we define a novel SW discrepancy, which we call spherical Sliced-Wasserstein, making a first step towards defining SW discrepancies on manifolds. Our construction is notably based on closed-form solutions of the Wasserstein distance on the circle, together with a new spherical Radon transform. Along with efficient algorithms and the corresponding implementations, we illustrate its properties in several machine learning use cases where spherical representations of data are at stake: sampling on the sphere, density estimation on real earth data or hyperspherical auto-encoders.

## 1 Introduction

Optimal transport (OT) (Villani, 2009) has received a lot of attention in machine learning in the past few years. As it allows to compare distributions with metrics, it has been used for different tasks such as domain adaptation (Courty et al., 2016) or generative models (Arjovsky et al., 2017), to name a few. The most classical distance used in OT is the Wasserstein distance. However, calculating it can be computationally expensive. Hence, several variants were proposed to alleviate the computational burden, such as the entropic regularization (Cuturi, 2013; Scetbon et al., 2021), minibatch OT (Fatras et al., 2020) or the sliced-Wasserstein distance (SW) for distributions supported on Euclidean spaces (Rabin et al., 2011b).

Although embedded in larger dimensional Euclidean spaces, data generally lie in practice on manifolds (Fefferman et al., 2016). A simple manifold, but with lots of practical applications, is the hypersphere $S^{d-1}$. Several types of data are by essence spherical: a good example is found in directional data (Mardia et al., 2000; Pewsey & García-Portugués, 2021) for which dedicated machine learning solutions are being developed (Sra, 2018), but other applications concern for instance geophysical data (Di Marzio et al., 2014), meteorology (Besombes et al., 2021), cosmology (Perraudin et al., 2019) or extreme value theory for the estimation of spectral measures (Guillou et al., 2015). Remarkably, in a more abstract setting, considering hyperspherical latent representations of data is becoming more and more common (*e.g.* Liu et al., 2017; Xu & Durrett, 2018; Davidson et al., 2018)). For example, in the context of variational autoencoders (Kingma & Welling, 2013), using priors on the sphere has been demonstrated to be beneficial (Davidson et al., 2018). Also, in the context of self-supervised learning (SSL), where one wants to learn discriminative representations in an unsupervised way, the hypersphere is usually considered for the latent representation (Wu et al., 2018; Chen et al., 2020a; Wang & Isola, 2020; Grill et al., 2020; Caron et al., 2020). It is thus of primary importance to develop machine learning tools that accommodate well with this specific geometry.

The OT theory on manifolds is well developed (Villani, 2009; Figalli & Villani, 2011; McCann, 2001) and several works started to use it in practice, with a focus mainly on the approximation of OT maps. For example, Cohen et al. (2021); Rezende & Racanière (2021) approximate the OT map to define normalizing flows on Riemannian manifolds, Hamfeldt & Turnquist (2021a;b); Cui et al. (2019) derive algorithms to approximate the OT map on the sphere, Alvarez-Melis et al. (2020); Hoyos-

Idrobo (2020) learn the transport map on hyperbolic spaces. However, the computational bottleneck to compute the Wasserstein distance on such spaces remains, and, as underlined in the conclusion of (Nadjahi, 2021), defining SW distances on manifolds would be of much interest. Notably, Rustamov & Majumdar (2020) proposed a variant of SW, based on the spectral decomposition of the Laplace-Beltrami operator, which generalizes to manifolds given the availability of the eigenvalues and eigenfunctions. However, it is not directly related to the original SW on Euclidean spaces.

**Contributions.** Therefore, by leveraging properties of the Wasserstein distance on the circle (Rabin et al., 2011a), we define the first, to the best of our knowledge, natural generalization of the original SW discrepancy on a non trivial manifold, namely the sphere $S^{d-1}$, and hence we make a first step towards defining SW distances on Riemannian manifolds. We make connections with a new spherical Radon transform and analyze some of its properties. We discuss the underlying algorithmic procedure, and notably provide an efficient implementation when computing the discrepancy against a uniform distribution. Then, we show that we can use this discrepancy on different tasks such as sampling, density estimation or generative modeling.

## 2 BACKGROUND

The aim of this paper is to define a Sliced-Wasserstein discrepancy on the hypersphere $S^{d-1} = \{x \in \mathbb{R}^d, \|x\|_2 = 1\}$. Therefore, in this section, we introduce the Wasserstein distance on manifolds and the classical SW distance on $\mathbb{R}^d$.

### 2.1 WASSERSTEIN DISTANCE

Since we are interested in defining a SW discrepancy on the sphere, we start by introducing the Wasserstein distance on a Riemannian manifold $M$ endowed with the Riemannian distance $d$. We refer to (Villani, 2009; Figalli & Villani, 2011) for more details.

Let $p \geq 1$ and $\mu, \nu \in \mathcal{P}_p(M) = \{\mu \in \mathcal{P}(M), \int_M d^p(x, x_0) \, d\mu(x) < \infty$ for some $x_0 \in M\}$. Then, the $p$-Wasserstein distance between $\mu$ and $\nu$ is defined as

$$W_p^p(\mu, \nu) = \inf_{\gamma \in \Pi(\mu,\nu)} \int_{M \times M} d^p(x, y) \, d\gamma(x, y), \tag{1}$$

where $\Pi(\mu, \nu) = \{\gamma \in \mathcal{P}(M \times M), \forall A \subset M, \gamma(M \times A) = \nu(A)$ and $\gamma(A \times M) = \mu(A)\}$ denotes the set of couplings.

For discrete probability measures, the Wasserstein distance can be computed using linear programs (Peyré et al., 2019). However, these algorithms have a $O(n^3 \log n)$ complexity *w.r.t.* the number of samples $n$ which is computationally intensive. Therefore, a whole literature consists of defining alternative discrepancies which are cheaper to compute. On Euclidean spaces, one of them is the Sliced-Wasserstein distance.

### 2.2 SLICED-WASSERSTEIN DISTANCE

On $M = \mathbb{R}^d$ with $d(x, y) = \|x - y\|_p^p$, a more attractive distance is the Sliced-Wasserstein (SW) distance. This distance relies on the appealing fact that for one dimensional measures $\mu, \nu \in \mathcal{P}(\mathbb{R})$, we have the following closed-form (Peyré et al., 2019, Remark 2.30)

$$W_p^p(\mu, \nu) = \int_0^1 \left| F_\mu^{-1}(u) - F_\nu^{-1}(u) \right|^p \, du, \tag{2}$$

where $F_\mu^{-1}$ (resp. $F_\nu^{-1}$) is the quantile function of $\mu$ (resp. $\nu$). From this property, Rabin et al. (2011b); Bonnotte (2013) defined the SW distance as

$$\forall \mu, \nu \in \mathcal{P}_p(\mathbb{R}^d), \ SW_p^p(\mu, \nu) = \int_{S^{d-1}} W_p^p(P_\#^\theta \mu, P_\#^\theta \nu) \, d\lambda(\theta), \tag{3}$$

where $P^\theta(x) = \langle x, \theta \rangle$, $\lambda$ is the uniform distribution on $S^{d-1}$ and for any Borel set $A \in \mathcal{B}(\mathbb{R}^d)$, $P_\#^\theta \mu(A) = \mu((P^\theta)^{-1}(A))$.

This distance can be approximated efficiently by using a Monte-Carlo approximation (Nadjahi et al., 2019), and amounts to a complexity of $O(Ln(d + \log n))$ where $L$ denotes the number of projections used for the Monte-Carlo approximation and $n$ the number of samples.

SW can also be written through the Radon transform (Bonneel et al., 2015). Let $f \in L^1(\mathbb{R}^d)$, then the Radon transform $R : L^1(\mathbb{R}^d) \to L^1(\mathbb{R} \times S^{d-1})$ is defined as (Helgason et al., 2011)

$$\forall \theta \in S^{d-1}, \ \forall t \in \mathbb{R}, \ Rf(t, \theta) = \int_{\mathbb{R}^d} f(x) \mathbb{1}_{\{\langle x, \theta \rangle = t\}} \mathrm{d}x. \tag{4}$$

Its dual $R^* : C_0(\mathbb{R} \times S^{d-1}) \to C_0(\mathbb{R}^d)$ (also known as back-projection operator), where $C_0$ denotes the set of continuous functions that vanish at infinity, satisfies for all $f, g$, $\langle Rf, g \rangle_{\mathbb{R} \times S^{d-1}} = \langle f, R^*g \rangle_{\mathbb{R}^d}$ and can be defined as (Boman & Lindskog, 2009; Bonneel et al., 2015)

$$\forall g \in C_0(\mathbb{R} \times S^{d-1}), \forall x \in \mathbb{R}^d, \ R^*g(x) = \int_{S^{d-1}} g(\langle x, \theta \rangle, \theta) \, \mathrm{d}\theta. \tag{5}$$

Therefore, by duality, we can define the Radon transform of a measure $\mu \in \mathcal{M}(\mathbb{R}^d)$ as the measure $R\mu \in \mathcal{M}(\mathbb{R} \times S^{d-1})$ such that for all $g \in C_0(\mathbb{R} \times S^{d-1})$, $\langle R\mu, g \rangle_{\mathbb{R} \times S^{d-1}} = \langle \mu, R^*g \rangle_{\mathbb{R}^d}$. Since $R\mu$ is a measure on the product space $\mathbb{R} \times S^{d-1}$, we can disintegrate it $w.r.t.$ $\lambda$, the uniform measure on $S^{d-1}$ (Ambrosio et al., 2005), as $R\mu = \lambda \otimes K$ with $K$ a probability kernel on $S^{d-1} \times \mathcal{B}(\mathbb{R})$, $i.e.$ for all $\theta \in S^{d-1}$, $K(\theta, \cdot)$ is a probability on $\mathbb{R}$, for any Borel set $A \in \mathcal{B}(\mathbb{R})$, $K(\cdot, A)$ is measurable, and

$$\forall \phi \in C(\mathbb{R} \times S^{d-1}), \ \int_{\mathbb{R} \times S^{d-1}} \phi(t, \theta) \mathrm{d}(R\mu)(t, \theta) = \int_{S^{d-1}} \int_{\mathbb{R}} \phi(t, \theta) K(\theta, \mathrm{d}t) \mathrm{d}\lambda(\theta), \tag{6}$$

with $C(\mathbb{R} \times S^{d-1})$ the set of continuous functions on $\mathbb{R} \times S^{d-1}$. By Proposition 6 in (Bonneel et al., 2015), we have that for $\lambda$-almost every $\theta \in S^{d-1}$, $(R\mu)^\theta = P^\theta_{\#}\mu$ where we denote $K(\theta, \cdot) = (R\mu)^\theta$. Therefore, we have

$$\forall \mu, \nu \in \mathcal{P}_p(\mathbb{R}^d), \ SW_p^p(\mu, \nu) = \int_{S^{d-1}} W_p^p\big((R\mu)^\theta, (R\mu)^\theta\big) \, \mathrm{d}\lambda(\theta). \tag{7}$$

Variants of SW have been defined in recent works, either by integrating $w.r.t.$ different distributions (Deshpande et al., 2019; Nguyen et al., 2021; 2020), by projecting on $\mathbb{R}$ using different projections (Nguyen & Ho, 2022a;b; Rustamov & Majumdar, 2020) or Radon transforms (Kolouri et al., 2019; Chen et al., 2020b), or by projecting on subspaces of higher dimensions (Paty & Cuturi, 2019; Lin et al., 2020; 2021; Huang et al., 2021).

## 3 A SLICED-WASSERSTEIN DISCREPANCY ON THE SPHERE

Our goal here is to define a sliced-Wasserstein distance on the sphere $S^{d-1}$. To that aim, we proceed analogously to the classical Euclidean space. We first rely on the nice properties of the Wasserstein distance on the circle (Rabin et al., 2011a) and then propose to project distributions lying on the sphere to great circles. Hence, circles play the role of the real line for the hypersphere. In this section, we first describe the OT problem on the circle, then we define a sliced-Wasserstein discrepancy on the sphere and discuss some of its properties. Notably, we derive a new spherical Radon transform which is linked to our newly defined spherical SW. We refer to Appendix A for the proofs.

### 3.1 OPTIMAL TRANSPORT ON THE CIRCLE

On the circle $S^1 = \mathbb{R}/\mathbb{Z}$ equipped with the geodesic distance $d_{S^1}$, an appealing formulation of the Wasserstein distance is available (Delon et al., 2010). First, let us parametrize $S^1$ by $[0, 1[$, then the geodesic distance can be written as (Rabin et al., 2011a), for all $x, y \in [0, 1[$, $d_{S^1}(x, y) = \min(|x - y|, 1 - |x - y|)$. Then, for the cost function $c(x, y) = h(d_{S^1}(x, y))$ with $h : \mathbb{R} \to \mathbb{R}^+$ an increasing convex function, the Wasserstein distance between $\mu \in \mathcal{P}(S^1)$ and $\nu \in \mathcal{P}(S^1)$ can be written as

$$W_c(\mu, \nu) = \inf_{\alpha \in \mathbb{R}} \int_0^1 h\big(|F_\mu^{-1}(t) - (F_\nu - \alpha)^{-1}(t)|\big) \, \mathrm{d}t, \tag{8}$$

where $F_\mu : [0, 1[ \to [0, 1]$ denotes the cumulative distribution function (cdf) of $\mu$, $F_\mu^{-1}$ its quantile function and $\alpha$ is a shift parameter. The optimization problem over the shifted cdf $F_\nu - \alpha$ can be seen

as looking for the best "cut" (or origin) of the circle into the real line because of the 1-periodicity. Indeed, the proof of this result for discrete distributions in (Rabin et al., 2011a) consists in cutting the circle at the optimal point and wrapping it around the real line, for which the optimal transport map is the increasing rearrangement $F_\nu^{-1} \circ F_\mu$ which can be obtained for discrete distributions by sorting the points (Peyré et al., 2019).

Rabin et al. (2011a) showed that the minimization problem is convex and coercive in the shift parameter and Delon et al. (2010) derived a binary search algorithm to find it. For the particular case of $h = \mathrm{Id}$, it can further be shown (Werman et al., 1985; Cabrelli & Molter, 1995) that

$$W_1(\mu, \nu) = \inf_{\alpha \in \mathbb{R}} \int_0^1 |F_\mu(t) - F_\nu(t) - \alpha| \, \mathrm{d}t. \tag{9}$$

In this case, we know exactly the minimum which is attained at the level median (Hundrieser et al., 2021). For $f : [0, 1[ \to \mathbb{R}$,

$$\mathrm{LevMed}(f) = \min \left\{ \operatorname*{argmin}_{\alpha \in \mathbb{R}} \int_0^1 |f(t) - \alpha| \mathrm{d}t \right\} = \inf \left\{ t \in \mathbb{R}, \ \beta(\{x \in [0, 1[, \ f(x) \le t\}) \ge \frac{1}{2} \right\}, \tag{10}$$

where $\beta$ is the Lebesgue measure. Therefore, we also have

$$W_1(\mu, \nu) = \int_0^1 |F_\mu(t) - F_\nu(t) - \mathrm{LevMed}(F_\mu - F_\nu)| \, \mathrm{d}t. \tag{11}$$

Since we know the minimum, we do not need the binary search and we can approximate the integral very efficiently as we only need to sort the samples to compute the level median and the cdfs.

Another interesting setting in practice is to compute $W_2$, *i.e.* with $h(x) = x^2$, *w.r.t.* a uniform distribution $\nu$ on the circle. We derive here the optimal shift $\hat{\alpha}$ for the Wasserstein distance between $\mu$ an arbitrary distribution on $S^1$ and $\nu$. We also provide a closed-form when $\mu$ is a discrete distribution.

**Proposition 1.** *Let $\mu \in \mathcal{P}_2(S^1)$ and $\nu = \mathrm{Unif}(S^1)$. Then,*

$$W_2^2(\mu, \nu) = \int_0^1 |F_\mu^{-1}(t) - t - \hat{\alpha}|^2 \, \mathrm{d}t \quad with \quad \hat{\alpha} = \int x \, \mathrm{d}\mu(x) - \frac{1}{2}. \tag{12}$$

*In particular, if $x_1 < \cdots < x_n$ and $\mu_n = \frac{1}{n} \sum_{i=1}^n \delta_{x_i}$, then*

$$W_2^2(\mu_n, \nu) = \frac{1}{n} \sum_{i=1}^n x_i^2 - \left( \frac{1}{n} \sum_{i=1}^n x_i \right)^2 + \frac{1}{n^2} \sum_{i=1}^n (n + 1 - 2i) x_i + \frac{1}{12}. \tag{13}$$

This proposition offers an intuitive interpretation: the optimal cut point between an empirical and a uniform distributions is the antipodal point of the circular mean of the discrete samples. Moreover, a very efficient algorithm can be derived from this property, as it solely requires a sorting operation to compute the order statistics of the samples.

## 3.2 DEFINITION OF SW ON THE SPHERE

On the hypersphere, the counterpart of straight lines are the great circles, which are circles with the same diameter as the sphere, and which correspond to the geodesics. Moreover, we can compute the Wasserstein distance on the circle fairly efficiently. Hence, to define a sliced-Wasserstein discrepancy on this manifold, we propose, analogously to the classical SW distance, to project measures on great circles. The most natural way to project points from $S^{d-1}$ to a great circle $C$ is to use the geodesic projection (Jung, 2021; Fletcher et al., 2004) defined as

$$\forall x \in S^{d-1}, \ P^C(x) = \operatorname*{argmin}_{y \in C} d_{S^{d-1}}(x, y), \tag{14}$$

where $d_{S^{d-1}}(x, y) = \arccos(\langle x, y \rangle)$ is the geodesic distance. See Figure 1 for an illustration of the geodesic projection on a great circle. Note that the projection is unique for almost every $x$ (see (Bardelli & Mennucci, 2017, Proposition 4.2) and Appendix B.1) and hence the pushforward $P_\#^C \mu$ of $\mu \in \mathcal{P}_{p,ac}(S^{d-1})$, where $\mathcal{P}_{p,ac}(S^{d-1})$ denotes the set of absolutely continuous measures *w.r.t.* the Lebesgue measure and with moments of order $p$, is well defined.

Great circles can be obtained by intersecting $S^{d-1}$ with a 2-dimensional plane (Jung et al., 2012). Therefore, to average over all great circles, we propose to integrate over the Grassmann manifold $\mathcal{G}_{d,2} = \{E \subset \mathbb{R}^d, \dim(E) = 2\}$ (Absil et al., 2004; Bendokat et al., 2020) and then to project the distribution onto the intersection with the hypersphere. Since the Grassmannian is not very practical, we consider the identification using the set of rank 2 projectors:

$$\mathcal{G}_{d,2} = \{P \in \mathbb{R}^{d \times d}, P^T = P, P^2 = P, \mathrm{Tr}(P) = 2\} = \{UU^T, U \in \mathbb{V}_{d,2}\}, \tag{15}$$

where $\mathbb{V}_{d,2} = \{U \in \mathbb{R}^{d \times 2}, U^T U = I_2\}$ is the Stiefel manifold (Bendokat et al., 2020).

Finally, we can define the Spherical Sliced-Wasserstein distance (SSW) for $p \geq 1$ between locally absolutely continuous measures *w.r.t.* the Lebesgue measure (Bardelli & Mennucci, 2017) $\mu, \nu \in \mathcal{P}_{p,\mathrm{ac}}(S^{d-1})$ as

$$SSW_p^p(\mu, \nu) = \int_{\mathbb{V}_{d,2}} W_p^p(P_\#^U \mu, P_\#^U \nu) \, \mathrm{d}\sigma(U), \tag{16}$$

where $\sigma$ is the uniform distribution over the Stiefel manifold $\mathbb{V}_{d,2}$, $P^U$ is the geodesic projection on the great circle generated by $U$ and then projected on $S^1$, *i.e.*

$$\forall U \in \mathbb{V}_{d,2}, \forall x \in S^{d-1}, \ P^U(x) = U^T \underset{y \in \mathrm{span}(UU^T) \cap S^{d-1}}{\mathrm{argmin}} d_{S^{d-1}}(x, y) = \underset{z \in S^1}{\mathrm{argmin}} \ d_{S^{d-1}}(x, Uz), \tag{17}$$

and the Wasserstein distance is defined with the geodesic distance $d_{S^1}$.

Moreover, we can derive a closed form expression which will be very useful in practice:

**Lemma 1.** *Let $U \in \mathbb{V}_{d,2}$ then for a.e. $x \in S^{d-1}$,*

$$P^U(x) = \frac{U^T x}{\|U^T x\|_2}. \tag{18}$$

Hence, we notice from this expression of the projection that we recover almost the same formula as Lin et al. (2020) but with an additional $\ell^2$ normalization which projects the data on the circle. As in (Lin et al., 2020), we could project on a higher dimensional subsphere by integrating over $\mathbb{V}_{d,k}$ with $k \geq 2$. However, we would lose the computational efficiency provided by the properties of the Wasserstein distance on the circle.

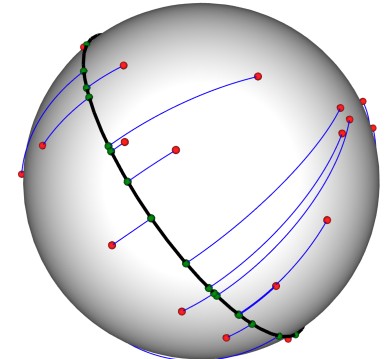

Figure 1: Illustration of the geodesic projections on a great circle (in black). In red, random points sampled on the sphere. In green the projections and in blue the trajectories.

### 3.3 A SPHERICAL RADON TRANSFORM

As for the classical SW distance, we can derive a second formulation using a Radon transform. Let $f \in L^1(S^{d-1})$, we define a spherical Radon transform $\tilde{R} : L^1(S^{d-1}) \to L^1(S^1 \times \mathbb{V}_{d,2})$ as

$$\forall z \in S^1, \ \forall U \in \mathbb{V}_{d,2}, \ \tilde{R}f(z, U) = \int_{S^{d-1}} f(x) \mathbb{1}_{\{z = P^U(x)\}} \mathrm{d}x. \tag{19}$$

This is basically the same formulation as the classical Radon transform (Natterer, 2001; Helgason et al., 2011) where we replaced the real line coordinate $t$ by the coordinate on the circle $z$ and the projection is the geodesic one which is well suited to the sphere. This transform is actually new since we integrate over different sets compared to existing works on spherical Radon transforms.

Then, analogously to the classical Radon transform, we can define the back-projection operator $\tilde{R}^* : C_0(S^1 \times \mathbb{V}_{d,2}) \to C_b(S^{d-1})$, $C_b(S^{d-1})$ being the space of continuous bounded functions, for $g \in C_0(S^1 \times \mathbb{V}_{d,2})$ as for a.e. $x \in S^{d-1}$,

$$\tilde{R}^* g(x) = \int_{\mathbb{V}_{d,2}} g(P^U(x), U) \, \mathrm{d}\sigma(U). \tag{20}$$

**Proposition 2.** *$\tilde{R}^*$ is the dual operator of $\tilde{R}$, i.e. for all $f \in L^1(S^{d-1})$, $g \in C_0(S^1 \times \mathbb{V}_{d,2})$,*

$$\langle \tilde{R}f, g \rangle_{S^1 \times \mathbb{V}_{d,2}} = \langle f, \tilde{R}^* g \rangle_{S^{d-1}}. \tag{21}$$

Now that we have a dual operator, we can also define the Radon transform of an absolutely continuous measure $\mu \in \mathcal{M}_{ac}(S^{d-1})$ by duality (Boman & Lindskog, 2009; Bonneel et al., 2015) as the measure $\tilde{R}\mu$ satisfying

$$\forall g \in C_0(S^1 \times \mathbb{V}_{d,2}), \int_{S^1 \times \mathbb{V}_{d,2}} g(z, U) \, \mathrm{d}(\tilde{R}\mu)(z, U) = \int_{S^{d-1}} \tilde{R}^* g(x) \, \mathrm{d}\mu(x). \qquad (22)$$

Since $\tilde{R}\mu$ is a measure on the product space $S^1 \times \mathbb{V}_{d,2}$, $\tilde{R}\mu$ can be disintegrated (Ambrosio et al., 2005, Theorem 5.3.1) *w.r.t.* $\sigma$ as $\tilde{R}\mu = \sigma \otimes K$ where $K$ is a probability kernel on $\mathbb{V}_{d,2} \times \mathcal{S}^1$ with $\mathcal{S}^1$ the Borel $\sigma$-field of $S^1$. We will denote for $\sigma$-almost every $U \in \mathbb{V}_{d,2}$, $(\tilde{R}\mu)^U = K(U, \cdot)$ the conditional probability.

**Proposition 3.** *Let* $\mu \in \mathcal{M}_{ac}(S^{d-1})$, *then for $\sigma$-almost every* $U \in \mathbb{V}_{d,2}$, $(\tilde{R}\mu)^U = P^U_\# \mu$.

Finally, we can write SSW (16) using this Radon transform:

$$\forall \mu, \nu \in \mathcal{P}_{p,ac}(S^{d-1}), \ SSW^p_p(\mu, \nu) = \int_{\mathbb{V}_{d,2}} W^p_p\big((\tilde{R}\mu)^U, (\tilde{R}\nu)^U\big) \, \mathrm{d}\sigma(U). \qquad (23)$$

Note that a natural way to define SW distances can be through already known Radon transforms using the formulation (23). It is for example what was done in (Kolouri et al., 2019) using generalized Radon transforms (Ehrenpreis, 2003; Homan & Zhou, 2017) to define generalized SW distances, or in (Chen et al., 2020b) with the spatial Radon transform. However, for known spherical Radon transforms (Abouelaz & Daher, 1993; Antipov et al., 2011) such as the Minkowski-Funk transform (Dann, 2010) or more generally the geodesic Radon transform (Rubin, 2002), there is no natural way that we know of to integrate over some product space and allowing to define a SW distance using disintegration.

As observed by Kolouri et al. (2019) for the generalized SW distances (GSW), studying the injectivity of the related Radon transforms allows to study the set on which SW is actually a distance. While the classical Radon transform integrates over hyperplanes of $\mathbb{R}^d$, the generalized Radon transform over hypersurfaces (Kolouri et al., 2019) and the Minkowski-Funk transform over "big circles", *i.e.* the intersection between a hyperplane and $S^{d-1}$ (Rubin, 2003), the set of integration here is a half of a big circle. Hence, $\tilde{R}$ is related to the hemispherical transform (Rubin, 1999) on $S^{d-2}$. We refer to Appendix A.6 for more details on the links with the hemispherical transform. Using these connections, we can derive the kernel of $\tilde{R}$ as the set of even measures which are null over all hyperplanes intersected with $S^{d-1}$.

**Proposition 4.** $\ker(\tilde{R}) = \{\mu \in \mathcal{M}_{\mathrm{even}}(S^{d-1}), \ \forall H \in \mathcal{G}_{d,d-1}, \ \mu(H \cap S^{d-1}) = 0\}$ *where* $\mu \in \mathcal{M}_{\mathrm{even}}$ *if for all* $f \in C(S^{d-1})$, $\langle \mu, f \rangle = \langle \mu, f_+ \rangle$ *with* $f_+(x) = (f(x) + f(-x))/2$ *for all* $x$.

We leave for future works checking whether this set is null or not. Hence, we conclude here that SSW is a pseudo-distance, but a distance on the sets of injectivity of $\tilde{R}$ (Agranovskyt & Quintott, 1996).

**Proposition 5.** *Let* $p \geq 1$, $SSW_p$ *is a pseudo-distance on* $\mathcal{P}_{p,ac}(S^{d-1})$.

## 4 IMPLEMENTATION

In practice, we approximate the distributions with empirical approximations and, as for the classical SW distance, we rely on the Monte-Carlo approximation of the integral on $\mathbb{V}_{d,2}$. We first need to sample from the uniform distribution $\sigma \in \mathcal{P}(\mathbb{V}_{d,2})$. This can be done by first constructing $Z \in \mathbb{R}^{d \times 2}$ by drawing each of its component from the standard normal distribution $\mathcal{N}(0, 1)$ and then applying the QR decomposition (Lin et al., 2021). Once we have $(U_\ell)_{\ell=1}^L \sim \sigma$, we project the samples on the circle $S^1$ by applying Lemma 1 and we compute the coordinates on the circle using the $\mathrm{atan2}$ function. Finally, we can compute the Wasserstein distance on the circle by either applying the binary search algorithm of (Delon et al., 2010) or the level median formulation (11) for $SSW_1$. In the particular case in which we want to compute $SSW_2$ between a measure $\mu$ and the uniform measure on the sphere $\nu = \mathrm{Unif}(S^{d-1})$, we can use the appealing fact that the projection of $\nu$ on the circle is uniform, *i.e.* $P^U_\# \nu = \mathrm{Unif}(S^1)$ (particular case of Theorem 3.1 in (Jung, 2021), see Appendix B.3). Hence, we can use the Proposition 1 to compute $W_2$, which allows a very efficient implementation either by the closed-form (13) or approximation by rectangle method of (12). This will be of particular interest for applications in Section 5 such as autoencoders. We sum up the procedure in Algorithm 1.

---

**Algorithm 1** SSW

---

**Input:** $(x_i)_{i=1}^n \sim \mu$, $(y_j)_{j=1}^m \sim \nu$, $L$ the number of projections, $p$ the order
**for** $\ell = 1$ **to** $L$ **do**
    Draw a random matrix $Z \in \mathbb{R}^{d \times 2}$ with for all $i, j$, $Z_{i,j} \sim \mathcal{N}(0, 1)$
    $U = \mathrm{QR}(Z) \sim \sigma$
    Project on $S^1$ the points: $\forall i, j$, $\hat{x}_i^\ell = \frac{U^T x_i}{\|U^T x_i\|_2}$, $\hat{y}_j^\ell = \frac{U^T y_j}{\|U^T y_j\|_2}$
    Compute the coordinates on the circle $S^1$: $\forall i, j$, $\tilde{x}_i^\ell = (\pi + \mathrm{atan2}(-x_{i,2}, -x_{i,1}))/(2\pi)$, $\tilde{y}_j^\ell = (\pi + \mathrm{atan2}(-y_{j,2}, -y_{j,1}))/(2\pi)$
    Compute $W_p^p(\frac{1}{n} \sum_{i=1}^n \delta_{\tilde{x}_i^\ell}, \frac{1}{m} \sum_{j=1}^m \delta_{\tilde{y}_j^\ell})$ by binary search or (11) for $p = 1$
**end for**
Return $SSW_p^p(\mu, \nu) \approx \frac{1}{L} \sum_{\ell=1}^L W_p^p(\frac{1}{n} \sum_{i=1}^n \delta_{\tilde{x}_i^\ell}, \frac{1}{m} \sum_{j=1}^m \delta_{\tilde{y}_j^\ell})$

---

**Complexity.** Let us note $n$ (resp. $m$) the number of samples of $\mu$ (resp. $\nu$), and $L$ the number of projections. First, we need to compute the QR factorization of $L$ matrices of size $d \times 2$. This can be done in $O(Ld)$ by using *e.g.* Householder reflections (Golub & Van Loan, 2013, Chapter 5.2) or the Scharwz-Rutishauser algorithm (Gander, 1980). Projecting the points on $S^1$ by Lemma 1 is in $O((n + m)dL)$ since we need to compute $L(n + m)$ products between $U_\ell^T \in \mathbb{R}^{2 \times d}$ and $x \in \mathbb{R}^d$. For the binary search or particular case formula (11) and (13), we need first to sort the points. But the binary search also adds a cost of $O((n + m) \log(\frac{1}{\epsilon}))$ to approximate the solution with precision $\epsilon$ (Delon et al., 2010) and the computation of the level median requires to sort $(n + m)$ points. Hence, for the general $SSW_p$, the complexity is $O(L(n + m)(d + \log(\frac{1}{\epsilon})) + Ln \log n + Lm \log m)$ versus $O(L(n + m)(d + \log(n + m)))$ for $SSW_1$ with the level median and $O(Ln(d + \log n))$ for $SSW_2$ against a uniform with the particular advantage that we do not need uniform samples in this case.

**Runtime Comparison.** We perform here some runtime comparisons. Using Pytorch (Paszke et al., 2019), we implemented the binary search algorithm of (Delon et al., 2010) and used it with $\epsilon = 10^{-6}$. We also implemented $SSW_1$ using the level median formula (11) and $SSW_2$ against a uniform measure (12). All experiments are conducted on GPU.

On Figure 2, we compare the runtime between two distributions on $S^2$ between SSW, SW, the Wasserstein distance and the entropic approximation using the Sinkhorn algorithm (Cuturi, 2013) with the geodesic distance as cost function. The distributions were approximated using $n \in \{10^2, 10^3, 10^4, 5 \cdot 10^4, 10^5, 5 \cdot 10^5\}$ samples of each distribution and we report the mean over 20 computations. We use the Python Optimal Transport (POT) library (Flamary et al., 2021) to compute the Wasserstein distance and the entropic approximation. For large enough batches, we observe that SSW is much faster than its Wasserstein counterpart, and it also scales better in term of memory because of the need to store the $n \times n$ cost matrix. For small batches, the computation of SSW actually takes longer because of the computation of the QR factorizations and of the projections. For bigger batches, it is bounded by the sorting operation

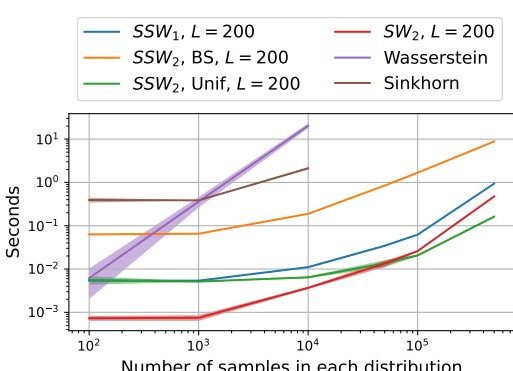

Figure 2: Runtime comparison in log-log scale between W, Sinkhorn with the geodesic distance, $SW_2$, $SSW_2$ with the binary search (BS) and uniform distribution (12) and $SSW_1$ with formula (11) between two distributions on $S^2$. The time includes the calculation of the distance matrices.

and we recover the quasi-linear slope. Furthermore, as expected, the fastest algorithms are $SSW_1$ with the level median and $SSW_2$ against a uniform as they have a quasilinear complexity. We report in Appendix C.2 other runtimes experiments *w.r.t.* to *e.g.* the number of projections or the dimension.

Additionally, we study both theoretically and empirically the projection and sample complexities in Appendices A.9 and C.1. We obtain similar results as (Nadjahi et al., 2020) derived for the SW distance. Notably, the sample complexity is independent *w.r.t.* the dimension.

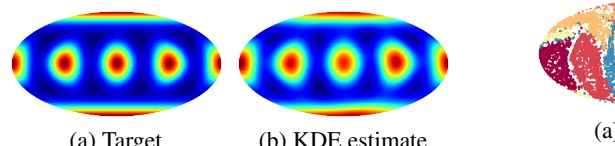

(a) Target      (b) KDE estimate

Figure 3: Minimization of SSW with respect to a mixture of vMF.

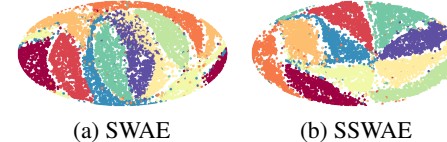

(a) SWAE      (b) SSWAE

Figure 4: Latent space of SWAE and SSWAE on MNIST for a uniform prior on $S^2$.

## 5 EXPERIMENTS

Apart from showing that SSW is an effective discrepancy for learning problems defined over the sphere, the objectives of this experimental Section is to show that it behaves better than using the more immediate SW in the embedding space. We first illustrate the ability to approximate different distributions by minimizing SSW *w.r.t.* some target distributions on $S^2$ and by performing density estimation experiments on real earth data. Then, we apply SSW for generative modeling tasks using the framework of Sliced-Wasserstein autoencoder and we show that we obtain competitive results with other Wasserstein autoencoder based methods using a prior on higher dimensional hyperspheres. Complete details about the experimental settings and optimization strategies are given in Appendix C. We also report in Appendices C.5 or C.7 complementary experiments on variational inference on the sphere or self-supervised learning with uniformity prior on the embedding hypersphere that further assess the effectiveness of SSW in a wide range of learning tasks. The code is available online[1].

### 5.1 SSW AS A LOSS

**Gradient flow on toy data.** We verify on the first experiments that we can learn some target distribution $\nu \in \mathcal{P}(S^{d-1})$ by minimizing SSW, *i.e.* we consider the minimization problem $\operatorname{argmin}_\mu SSW_p^p(\mu, \nu)$. We suppose that we have access to the target distribution $\nu$ through samples, *i.e.* through $\hat{\nu}_m = \frac{1}{m} \sum_{j=1}^m \delta_{y_j}$ where $(y_j)_{j=1}^m$ are i.i.d samples of $\nu$. We add in Appendix C.5 the case where we know the density up to some constant which can be dealt with the sliced-Wasserstein variational inference framework introduced in (Yi & Liu, 2021). We choose as target distribution a mixture of 6 well separated von Mises-Fisher distributions (Mardia, 1975). This is a fairly challenging distribution since there are 6 modes which are not connected. We show on Figure 3 the Mollweide projection of the density approximated by a kernel density estimator for a distribution with 500 particles. To optimize directly over particles, we perform a Riemannian gradient descent on the sphere (Absil et al., 2009).

**Density estimation on earth data.** We perform density estimation on datasets first gathered by Mathieu & Nickel (2020) which contain locations of wild fires (EOSDIS, 2020), floods (Brakenridge, 2017) or eathquakes (NOAA, 2022). We use exponential map normalizing flows introduced in (Rezende et al., 2020) (see Appendix B.4) which are invertible transformations mapping the data to some prior that we need to enforce. Here, we choose as prior a uniform distribution on $S^2$ and we learn the model using SSW. These transformations allow to evaluate exactly the density at any point. More precisely, let $T$ be such transformation, let $p_Z$ be a prior distribution on $S^2$ and $\mu$ the measure of interest, which we know from samples, *i.e.* through $\hat{\mu}_n = \frac{1}{n} \sum_{i=1}^n \delta_{x_i}$. Then, we solve the following optimization problem $\min_T SSW_2^2(T_\# \mu, p_Z)$. Once it is fitted, then the learned density $f_\mu$ can be obtained by

$$\forall x \in S^2, \ f_\mu(x) = p_Z(T(x))|\det J_T(x)|, \tag{24}$$

where we used the change of variable formula.

Table 1: Negative test log likelihood.

|        | Earthquake | Flood | Fire |
|--------|------------|-------|------|
| SSW | $\mathbf{0.84}_{\pm 0.07}$ | $\mathbf{1.26}_{\pm 0.05}$ | $\mathbf{0.23}_{\pm 0.18}$ |
| SW | $0.94_{\pm 0.02}$ | $1.36_{\pm 0.04}$ | $0.54_{\pm 0.37}$ |
| Stereo | $1.91_{\pm 0.1}$ | $2.00_{\pm 0.07}$ | $1.27_{\pm 0.09}$ |

We show on Figure 5 the density of test data learned. We observe on this figure that the normalizing flows (NFs) put mass where most data points lie, and hence are able to somewhat recover the principle

---

[1]https://github.com/clbonet/Spherical_Sliced-Wasserstein

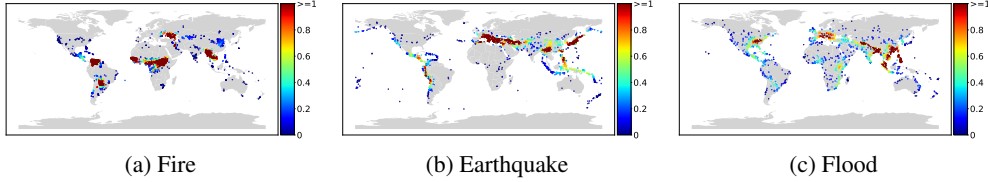

| (a) Fire | (b) Earthquake | (c) Flood |

Figure 5: Density estimation of models trained on earth data. We plot the density on the test data.

modes of the data.We also compare on Table 1 the negative test log likelihood, averaged over 5 trainings with different split of the data, between different OT metrics, namely SSW, SW and the stereographic projection model (Gemici et al., 2016) which first projects the data on $\mathbb{R}^2$ and use a regular NF in the projected space. We observe that SSW allows to better fit the data compared to the other OT based methods which are less suited to the sphere.

## 5.2 SSW AUTOENCODERS

Table 2: FID (Lower is better).

| Method / Dataset | MNIST | Fashion | CIFAR10 |
|---|---|---|---|
| SSWAE | $\mathbf{14.91}_{\pm 0.32}$ | $\mathbf{43.94}_{\pm 0.81}$ | $98.57_{\pm 035}$ |
| SWAE | $15.18_{\pm 0.32}$ | $44.78_{\pm 1.07}$ | $\mathbf{98.5}_{\pm \mathbf{0.45}}$ |
| WAE-MMD IMQ | $18.12_{\pm 0.62}$ | $68.51_{\pm 2.76}$ | $100.14_{\pm 0.67}$ |
| WAE-MMD RBF | $20.09_{\pm 1.42}$ | $70.58_{\pm 1.75}$ | $100.27_{\pm 0.74}$ |
| SAE | $19.39_{\pm 0.56}$ | $56.75_{\pm 1.7}$ | $99.34_{\pm 0.96}$ |
| Circular GSWAE | $15.01_{\pm 0.26}$ | $44.65_{\pm 1.2}$ | $98.8_{\pm 0.68}$ |

In this section, we use SSW to learn the latent space of autoencoders (AE). We rely on the SWAE framework introduced by Kolouri et al. (2018). Let $f$ be some encoder and $g$ be some decoder, denote $p_Z$ a prior distribution, then the loss minimized in SWAE is

$$\mathcal{L}(f, g) = \int c\big(x, g(f(x))\big) \mathrm{d}\mu(x) + \lambda SW_2^2(f_\# \mu, p_Z), \qquad (25)$$

where $\mu$ is the distribution of the data for which we have access to samples. One advantage of this framework over more classical VAEs (Kingma & Welling, 2013) is that no parametrization trick is needed here and therefore the choice of the prior is more free.

In several concomitant works, it was shown that using a prior on the hypersphere can improve the results (Davidson et al., 2018; Xu & Durrett, 2018). Hence, we propose in the same fashion as (Kolouri et al., 2018; 2019; Patrini et al., 2020) to replace SW by SSW, which we denote SSWAE, and to enforce a prior on the sphere. In the following, we use the MNIST (LeCun & Cortes, 2010), FashionMNIST (Xiao et al., 2017) and CIFAR10 (Krizhevsky, 2009) datasets, and we put an $\ell^2$ normalization at the output of the encoder. As a prior, we use the uniform distribution on $S^{10}$ for MNIST and Fashion, and on $S^{64}$ for CIFAR10. We compare in Table 2 the Fréchet Inception Distance (FID) (Heusel et al., 2017), for 10000 samples and averaged over 5 trainings, obtained with the Wasserstein Autoencoder (WAE) (Tolstikhin et al., 2018), the classical SWAE (Kolouri et al., 2018), the Sinkhorn Autoencoder (SAE) (Patrini et al., 2020) and circular GSWAE (Kolouri et al., 2019). We observe that we obtain fairly competitive results on the different datasets. We add on Figure 4 the latent space obtained with a uniform prior on $S^2$ on MNIST. We notably observe a better separation between classes for SSWAE.

## 6 CONCLUSION AND DISCUSSION

In this work, we derive a new sliced-Wasserstein discrepancy on the hypersphere, that comes with practical advantages when computing optimal transport distances on hyperspherical data. We notably showed that it is competitive or even sometimes better than other metrics defined directly on $\mathbb{R}^d$ on a variety of machine learning tasks, including density estimation or generative models. Our work is, up to our knowledge, the first to adapt the classical sliced Wasserstein framework to non-trivial manifolds. The three main ingredients are: *i)* a closed-form for Wasserstein on the circle, *ii)* a closed-form solution to the projection onto great circles, and *iii)* a novel Radon transform on the Sphere. An immediate extension of this work would be to consider sliced-Wasserstein discrepancy in hyperbolic spaces, where geodesics are circular arcs as in the Poincaré disk. Beyond the generalization to other, possibly well behaved, manifolds, asymptotic properties as well as statistical and topological aspects need to be examined. While we postulate that results comparable to the Euclidean case might be reached, the fact that the manifold is closed might bring interesting differences and justify further use of this type of discrepancies rather than their Euclidean counterparts.

ACKNOWLEDGMENTS

Clément Bonet thanks Benoît Malézieux for fruitful discussions. This work was performed partly using HPC resources from GENCI-IDRIS (Grant 2022-AD011013514). This research was funded by project DynaLearn from Labex CominLabs and Région Bretagne ARED DLearnMe, and by the project OTTOPIA ANR-20-CHIA-0030 of the French National Research Agency (ANR).

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

## A   PROOFS

### A.1   PROOF OF PROPOSITION 1

**Optimal $\alpha$.**   Let $\mu \in \mathcal{P}_2(S^1)$, $\nu = \text{Unif}(S^1)$. Since $\nu$ is the uniform distribution on $S^1$, its cdf is the identity on $[0, 1]$ (where we identified $S^1$ and $[0, 1]$). We can extend the cdf $F$ on the real line as in (Rabin et al., 2011a) with the convention $F(y + 1) = F(y) + 1$. Therefore, $F_\nu = \text{Id}$ on $\mathbb{R}$. Moreover, we know that for all $x \in S^1$, $(F_\nu - \alpha)^{-1}(x) = F_\nu^{-1}(x + \alpha) = x + \alpha$ and

$$W_2^2(\mu, \nu) = \inf_{\alpha \in \mathbb{R}} \int_0^1 |F_\mu^{-1}(t) - (F_\nu - \alpha)^{-1}(t)|^2 \, \mathrm{d}t. \tag{26}$$

For all $\alpha \in \mathbb{R}$, let $f(\alpha) = \int_0^1 \left( F_\mu^{-1}(t) - (F_\nu - \alpha)^{-1}(t) \right)^2 \mathrm{d}t$. Then, we have:

$$
\begin{aligned}
\forall \alpha \in \mathbb{R}, \ f(\alpha) &= \int_0^1 \left( F_\mu^{-1}(t) - t - \alpha \right)^2 \mathrm{d}t \\
&= \int_0^1 \left( F_\mu^{-1}(t) - t \right)^2 \mathrm{d}t + \alpha^2 - 2\alpha \int_0^1 (F_\mu^{-1}(t) - t) \, \mathrm{d}t \\
&= \int_0^1 \left( F_\mu^{-1}(t) - t \right)^2 \mathrm{d}t + \alpha^2 - 2\alpha \left( \int_0^1 x \, \mathrm{d}\mu(x) - \frac{1}{2} \right),
\end{aligned}
\tag{27}
$$

where we used that $(F_\mu^{-1})_\# \text{Unif}([0, 1]) = \mu$.

Hence, $f'(\alpha) = 0 \iff \alpha = \int_0^1 x \, \mathrm{d}\mu(x) - \frac{1}{2}$.

**Closed-form for empirical distributions.**   Let $(x_i)_{i=1}^n \in [0, 1[^n$ such that $x_1 < \cdots < x_n$ and let $\mu_n = \frac{1}{n} \sum_{i=1}^n \delta_{x_i}$ a discrete distribution.

To compute the closed-form of $W_2$ between $\mu_n$ and $\nu = \text{Unif}(S^1)$, we first have that the optimal $\alpha$ is $\alpha_n = \frac{1}{n} \sum_{i=1}^n x_i - \frac{1}{2}$. Moreover, we also have:

$$
\begin{aligned}
W_2^2(\mu_n, \nu) &= \int_0^1 \left( F_{\mu_n}^{-1}(t) - (t + \hat{\alpha}_n) \right)^2 \mathrm{d}t \\
&= \int_0^1 F_{\mu_n}^{-1}(t)^2 \, \mathrm{d}t - 2 \int_0^1 t F_{\mu_n}^{-1}(t) \mathrm{d}t - 2\hat{\alpha}_n \int_0^1 F_{\mu_n}^{-1}(t) \mathrm{d}t + \frac{1}{3} + \hat{\alpha}_n + \hat{\alpha}_n^2.
\end{aligned}
\tag{28}
$$

Then, by noticing that $F_{\mu_n}^{-1}(t) = x_i$ for all $t \in [F(x_i), F(x_{i+1})[$, we have

$$\int_0^1 t F_{\mu_n}^{-1}(t) \mathrm{d}t = \sum_{i=1}^n \int_{\frac{i-1}{n}}^{\frac{i}{n}} t x_i \mathrm{d}t = \frac{1}{2n^2} \sum_{i=1}^n x_i(2i - 1), \tag{29}$$

$$\int_0^1 F_\mu^{-1}(t)^2 \mathrm{d}t = \frac{1}{n} \sum_{i=1}^n x_i^2, \quad \int_0^1 F_\mu^{-1}(t) \mathrm{d}t = \frac{1}{n} \sum_{i=1}^n x_i, \tag{30}$$

and we also have:

$$\hat{\alpha}_n + \hat{\alpha}_n^2 = \frac{1}{n} \sum_{i=1}^n x_i - \frac{1}{2} + \left( \frac{1}{n} \sum_{i=1}^n x_i \right)^2 + \frac{1}{4} - \frac{1}{n} \sum_{i=1}^n x_i = \left( \frac{1}{n} \sum_{i=1}^n x_i \right)^2 - \frac{1}{4}. \tag{31}$$

Then, by plugging these results into (28), we obtain

$$
\begin{aligned}
W_2^2(\mu_n, \nu) &= \frac{1}{n} \sum_{i=1}^n x_i^2 - \frac{1}{n^2} \sum_{i=1}^n (2i - 1)x_i - 2\left( \frac{1}{n} \sum_{i=1}^n x_i \right)^2 + \frac{1}{n} \sum_{i=1}^n x_i + \frac{1}{3} + \left( \frac{1}{n} \sum_{i=1}^n x_i \right)^2 - \frac{1}{4} \\
&= \frac{1}{n} \sum_{i=1}^n x_i^2 - \left( \frac{1}{n} \sum_{i=1}^n x_i \right)^2 + \frac{1}{n^2} \sum_{i=1}^n (n + 1 - 2i)x_i + \frac{1}{12}.
\end{aligned}
\tag{32}
$$

## A.2    PROOF OF EQUATION 17

Let $U \in \mathbb{V}_{d,2}$. Then the great circle generated by $U \in \mathbb{V}_{d,2}$ is defined as the intersection between $\mathrm{span}(UU^T)$ and $S^{d-1}$. And we have the following characterization:

$$
\begin{aligned}
x \in \mathrm{span}(UU^T) \cap S^{d-1} &\iff \exists y \in \mathbb{R}^d, \ x = UU^T y \text{ and } \|x\|_2^2 = 1 \\
&\iff \exists y \in \mathbb{R}^d, \ x = UU^T y \text{ and } \|UU^T y\|_2^2 = y^T UU^T y = \|U^T y\|_2^2 = 1 \\
&\iff \exists z \in S^1, \ x = Uz.
\end{aligned}
$$

And we deduce that

$$
\forall U \in \mathbb{V}_{d,2}, x \in S^{d-1}, \ P^U(x) = \underset{z \in S^1}{\mathrm{argmin}} \ d_{S^{d-1}}(x, Uz). \tag{33}
$$

## A.3    PROOF OF LEMMA 1

Let $U \in \mathbb{V}_{d,2}$ and $x \in S^{d-1}$ such that $U^T x \neq 0$. Denote $U = (u_1 \ u_2)$, *i.e.* the 2-plane $E$ is $E = \mathrm{span}(UU^T) = \mathrm{span}(u_1, u_2)$ and $(u_1, u_2)$ is an orthonormal basis of $E$. Then, for all $x \in S^{d-1}$, the projection on $E$ is $p^E(x) = \langle u_1, x \rangle u_1 + \langle u_2, x \rangle u_2 = UU^T x$.

Now, let us compute the geodesic distance between $x \in S^{d-1}$ and $\frac{p^E(x)}{\|p^E(x)\|_2} \in E \cap S^{d-1}$:

$$
d_{S^{d-1}}\left(x, \frac{p^E(x)}{\|p^E(x)\|_2}\right) = \arccos\left(\langle x, \frac{p^E(x)}{\|p^E(x)\|_2}\rangle\right) = \arccos(\|p^E(x)\|_2), \tag{34}
$$

using that $x = p^E(x) + p^{E^\perp}(x)$.

Let $y \in E \cap S^{d-1}$ another point on the great circle. By the Cauchy-Schwarz inequality, we have

$$
\langle x, y \rangle = \langle p^E(x), y \rangle \leq \|p^E(x)\|_2 \|y\|_2 = \|p^E(x)\|_2. \tag{35}
$$

Therefore, using that $\arccos$ is decreasing on $(-1, 1)$,

$$
d_{S^{d-1}}(x, y) = \arccos(\langle x, y \rangle) \geq \arccos(\|p^E(x)\|_2) = d_{S^{d-1}}\left(x, \frac{p^E(x)}{\|p^E(x)\|_2}\right). \tag{36}
$$

Moreover, we have equality if and only if $y = \lambda p^E(x)$. And since $y \in S^{d-1}$, $|\lambda| = \frac{1}{\|p^E(x)\|_2}$. Using again that $\arccos$ is decreasing, we deduce that the minimum is well attained in $y = \frac{p^E(x)}{\|p^E(x)\|_2} = \frac{UU^T x}{\|UU^T x\|_2}$.

Finally, using that $\|UU^T x\|_2 = x^T UU^T UU^T x = x^T UU^T x = \|U^T x\|_2$, we deduce that

$$
P^U(x) = \frac{U^T x}{\|U^T x\|_2}. \tag{37}
$$

Finally, by noticing that the projection is unique if and only if $U^T x = 0$, and using (Bardelli & Mennucci, 2017, Proposition 4.2) which states that there is a unique projection for a.e. $x$, we deduce that $\{x \in S^{d-1}, \ U^T x = 0\}$ is of measure null and hence, for a.e. $x \in S^{d-1}$, we have the result.

## A.4 Proof of Proposition 2

Let $f \in L^1(S^{d-1})$, $g \in C_0(S^1 \times \mathbb{V}_{d,2})$, then by Fubini's theorem,

$$
\begin{aligned}
\langle \tilde{R}f, g \rangle_{S^1 \times \mathbb{V}_{d,2}} &= \int_{V_{d,2}} \int_{S^1} \tilde{R}f(z,U)g(z,U) \, \mathrm{d}z \mathrm{d}\sigma(U) \\
&= \int_{V_{d,2}} \int_{S^1} \int_{S^{d-1}} f(x) \mathbb{1}_{\{z = P^U(x)\}} g(z,U) \, \mathrm{d}x \mathrm{d}z \mathrm{d}\sigma(U) \\
&= \int_{S^{d-1}} f(x) \int_{V_{d,2}} \int_{S^1} g(z,U) \mathbb{1}_{\{z = P^U(x)\}} \, \mathrm{d}z \mathrm{d}\sigma(U) \mathrm{d}x \\
&= \int_{S^{d-1}} f(x) \int_{V_{d,2}} g(P^U(x), U) \, \mathrm{d}\sigma(U) \mathrm{d}x \\
&= \int_{S^{d-1}} f(x) \tilde{R}^* g(x) \, \mathrm{d}x \\
&= \langle f, \tilde{R}^* g \rangle_{S^{d-1}}.
\end{aligned}
\tag{38}
$$

## A.5 Proof of Proposition 3

Let $g \in C_0(S^1 \times \mathbb{V}_{d,2})$,

$$
\begin{aligned}
\int_{\mathbb{V}_{d,2}} \int_{S^1} g(z,U) \, (\tilde{R}\mu)^U(\mathrm{d}z) \, \mathrm{d}\sigma(U) &= \int_{S^1 \times \mathbb{V}_{d,2}} g(z,U) \, \mathrm{d}(\tilde{R}\mu)(z,U) \\
&= \int_{S^{d-1}} \tilde{R}^* g(x) \, \mathrm{d}\mu(x) \\
&= \int_{S^{d-1}} \int_{\mathbb{V}_{d,2}} g(P^U(x), U) \, \mathrm{d}\sigma(U) \mathrm{d}\mu(x) \\
&= \int_{\mathbb{V}_{d,2}} \int_{S^{d-1}} g(P^U(x), U) \, \mathrm{d}\mu(x) \mathrm{d}\sigma(U) \\
&= \int_{\mathbb{V}_{d,2}} \int_{S^1} g(z,U) \, \mathrm{d}(P^U_{\#}\mu)(z) \mathrm{d}\sigma(U).
\end{aligned}
\tag{39}
$$

Hence, for $\sigma$-almost every $U \in \mathbb{V}_{d,2}$, $(\tilde{R}\mu)^U = P^U_{\#}\mu$.

## A.6 Study of the Spherical Radon transform $\tilde{R}$

In this Section, we first discuss the set of integration of the spherical Radon transform $\tilde{R}$ (19). We further show that it is related to the hemispherical Radon transform and we derive its kernel.

**Set of integration.** While the classical Radon transform integrates over hyperplanes of $\mathbb{R}^d$ and the generalized Radon transform integrates over hypersurfaces (Kolouri et al., 2019), the set of integration of the spherical Radon transform (19) is a half of a "big circle", *i.e.* half of the intersection between a hyperplane and $S^{d-1}$ (Rubin, 2003). We illustrate this on $S^2$ in Figure 6. On $S^2$, the intersection between a hyperplane and $S^2$ is a great circle.

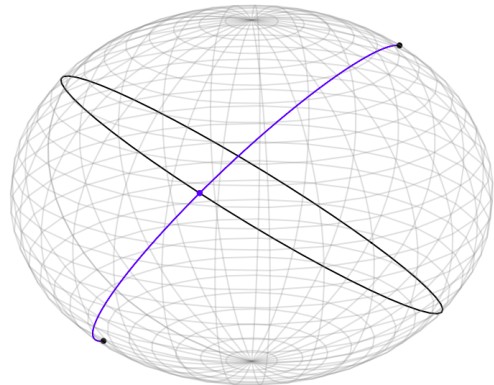

Figure 6: Set of integration of the spherical Radon transform (19). The great circle is in black and the set of integration in blue. The point $Uz \in \mathrm{span}(UU^T) \cap S^{d-1}$ is in blue.

**Proposition 6.** *Let $U \in \mathbb{V}_{d,2}$, $z \in S^1$. The set of integration of (19) is*

$$\{x \in S^{d-1}, \ P^U(x) = z\} = \{x \in F \cap S^{d-1}, \ \langle x, Uz \rangle > 0\}, \tag{40}$$

*where $F = \mathrm{span}(UU^T)^{\perp} \oplus \mathrm{span}(Uz)$.*

*Proof.* Let $U \in \mathbb{V}_{d,2}$, $z \in S^1$. Denote $E = \mathrm{span}(UU^T)$ the 2-plane generating the great circle, and $E^{\perp}$ its orthogonal complementary. Hence, $E \oplus E^{\perp} = \mathbb{R}^d$ and $\dim(E^{\perp}) = d - 2$. Now, let $F = E^{\perp} \oplus \mathrm{span}(Uz)$. Since $Uz = UU^T Uz \in E$, we have that $\dim(F) = d - 1$. Hence, $F$ is a hyperplane and $F \cap S^{d-1}$ is a "big circle" (Rubin, 2003), *i.e.* a $(d-2)$-dimensional subsphere of $S^{d-1}$.

Now, for the first inclusion, let $x \in \{x \in S^{d-1}, \ P^U(x) = z\}$. First, we show that $x \in F \cap S^{d-1}$. By Lemma 1 and hypothesis, we know that $P^U(x) = \frac{U^T x}{\|U^T x\|_2} = z$. By denoting by $p^E$ the projection on $E$, we have:

$$p^E(x) = UU^T x = U(\|U^T x\|_2 z) = \|U^T x\|_2 Uz \in \mathrm{span}(Uz). \tag{41}$$

Hence, $x = p^E(x) + x_{E^{\perp}} = \|U^T x\|_2 Uz + x_{E^{\perp}} \in F$. Moreover, as

$$\langle x, Uz \rangle = \|U^T x\|_2 \langle Uz, Uz \rangle = \|U^T x\|_2 > 0, \tag{42}$$

we deduce that $x \in \{F \cap S^{d-1}, \ \langle x, Uz \rangle > 0\}$.

For the other inclusion, let $x \in \{F \cap S^{d-1}, \ \langle x, Uz \rangle > 0\}$. Since $x \in F$, we have $x = x_{E^{\perp}} + \lambda Uz$, $\lambda \in \mathbb{R}$. Hence, using Lemma 1,

$$P^U(x) = \frac{U^T x}{\|U^T x\|_2} = \frac{\lambda}{|\lambda|} \frac{z}{\|z\|_2} = \mathrm{sign}(\lambda) z. \tag{43}$$

But, we also have $\langle x, Uz \rangle = \lambda \|Uz\|_2^2 = \lambda > 0$. Therefore, $\mathrm{sign}(\lambda) = 1$ and $P^U(x) = z$.

Finally, we conclude that $\{x \in S^{d-1}, \ P^U(x) = z)\} = \{x \in F \cap S^{d-1}, \ \langle x, Uz \rangle > 0\}$. $\qquad \square$

**Link with Hemispherical transform.** Since the intersection between a hyperplane and $S^{d-1}$ is isometric to $S^{d-2}$ (Jung et al., 2012), we can relate $\tilde{R}$ to the hemispherical transform $\mathcal{H}$ (Rubin, 2003) on $S^{d-2}$. First, the hemispherical transform of a function $f \in L^1(S^{d-1})$ is defined as

$$\forall x \in S^{d-1}, \ \mathcal{H}f(x) = \int_{S^{d-1}} f(y) \mathbb{1}_{\{\langle x, y \rangle > 0\}} \mathrm{d}y. \tag{44}$$

From Proposition 6, we can write the spherical Radon transform (19) as a hemispherical transform on $S^{d-2}$.

**Proposition 7.** *Let $f \in L^1(S^{d-1})$, $U \in \mathbb{V}_{d,2}$ and $z \in S^1$, then*

$$\tilde{R}f(z,U) = \int_{S^{d-2}} \tilde{f}(x)\mathbb{1}_{\{\langle x,\tilde{U}z\rangle>0\}}\mathrm{d}x = \mathcal{H}\tilde{f}(\tilde{U}z), \tag{45}$$

*where for all $x \in S^{d-2}$, $\tilde{f}(x) = f(O^TJx)$ with $O$ the rotation matrix such that for all $x \in F$, $Ox \in \mathrm{span}(e_1,\ldots,e_{d-1})$ where $(e_1,\ldots,e_d)$ denotes the canonical basis, and $J = \begin{pmatrix} I_{d-1} \\ 0_{1,d-1} \end{pmatrix}$, and $\tilde{U} = J^TOU \in \mathbb{R}^{(d-1)\times 2}$.*

*Proof.* Let $f \in L^1(S^{d-1})$, $z \in S^1$, $U \in \mathbb{V}_{d,2}$, then by Proposition 6,

$$\tilde{R}f(z,U) = \int_{S^{d-1}\cap F} f(x)\mathbb{1}_{\{\langle x,Uz\rangle>0\}}\mathrm{d}x. \tag{46}$$

$F$ is a hyperplane. Let $O \in \mathbb{R}^{d\times d}$ be the rotation such that for all $x \in F$, $Ox \in \mathrm{span}(e_1,\ldots,e_{d-1}) = \tilde{F}$ where $(e_1,\ldots,e_d)$ is the canonical basis. By applying the change of variable $Ox = y$, and since $O^{-1} = O^T$, $\det O = 1$, we obtain

$$\tilde{R}f(z,U) = \int_{O(F\cap S^{d-1})} f(O^Ty)\mathbb{1}_{\{\langle O^Ty,Uz\rangle>0\}}\mathrm{d}y = \int_{\tilde{F}\cap S^{d-1}} f(O^Ty)\mathbb{1}_{\{\langle y,OUz\rangle>0\}}\mathrm{d}y. \tag{47}$$

Now, we have that $OU \in \mathbb{V}_{d,2}$ since $(OU)^T(OU) = I_2$, and since $Uz \in F$, $OUz \in \tilde{F}$. For all $y \in \tilde{F}$, we have $\langle y,e_d\rangle = y_d = 0$. Let $J = \begin{pmatrix} I_{d-1} \\ 0_{1,d-1} \end{pmatrix} \in \mathbb{R}^{d\times(d-1)}$, then for all $y \in \tilde{F}\cap S^{d-1}$, $y = J\tilde{y}$ where $\tilde{y} \in S^{d-2}$ is composed of the $d-1$ first coordinates of $y$.

Let's define, for all $\tilde{y} \in S^{d-2}$, $\tilde{f}(\tilde{y}) = f(O^TJ\tilde{y})$, $\tilde{U} = J^TOU$.

Then, since $\tilde{F}\cap S^{d-1} \cong S^{d-2}$, we can write:

$$\tilde{R}f(z,U) = \int_{S^{d-2}} \tilde{f}(\tilde{y})\mathbb{1}_{\{\langle\tilde{y},\tilde{U}z\rangle>0\}}\mathrm{d}\tilde{y} = \mathcal{H}\tilde{f}(\tilde{U}z). \tag{48}$$

$\square$

**Kernel of $\tilde{R}$.** By exploiting the expression using the hemispherical transform in Proposition 7, we can derive its kernel in Appendix A.7.

## A.7    Proof of Proposition 4

First, we recall Lemma 2.3 of (Rubin, 1999) on $S^{d-2}$.

**Lemma 2** (Lemma 2.3 (Rubin, 1999)). $\ker(\mathcal{H}) = \{\mu \in \mathcal{M}_{\mathrm{even}}(S^{d-2}), \mu(S^{d-2}) = 0\}$ *where $\mathcal{M}_{\mathrm{even}}$ is the set of even measures, i.e. measures such that for all $f \in C(S^{d-2})$, $\langle\mu,f\rangle = \langle\mu,f^-\rangle$ where $f^-(x) = f(-x)$ for all $x \in S^{d-2}$.*

Let $\mu \in \mathcal{M}_{ac}(S^{d-1})$. First, we notice that the density of $\tilde{R}\mu$ w.r.t. $\lambda\otimes\sigma$ is, for all $z \in S^1$, $U \in \mathbb{V}_{d,2}$,

$$(\tilde{R}\mu)(z,U) = \int_{S^{d-1}} \mathbb{1}_{\{P^U(x)=z\}}\mathrm{d}\mu(x) = \int_{F\cap S^{d-1}} \mathbb{1}_{\{\langle x,Uz\rangle>0\}}\mathrm{d}\mu(x). \tag{49}$$

Indeed, using Proposition 2, and Proposition 6, we have for all $g \in C_0(S^1\times\mathbb{V}_{d,2})$,

$$\begin{aligned}
\langle\tilde{R}\mu,g\rangle_{S^1\times\mathbb{V}_{d,2}} = \langle\mu,\tilde{R}^*g\rangle_{S^{d-1}} &= \int_{S^{d-1}} R^*g(x)\mathrm{d}\mu(x) \\
&= \int_{S^{d-1}}\int_{\mathbb{V}_{d,2}}\int_{S^1} g(z,U)\mathbb{1}_{\{z=P^U(x)\}}\mathrm{d}z\mathrm{d}\sigma(U)\mathrm{d}\mu(x) \\
&= \int_{\mathbb{V}_{d,2}\times S^1} g(z,U)\int_{S^{d-1}}\mathbb{1}_{\{z=P^U(x)\}}\mathrm{d}\mu(x)\ \mathrm{d}z\mathrm{d}\sigma(U) \\
&= \int_{\mathbb{V}_{d,2}\times S^1} g(z,U)\int_{F\cap S^{d-1}}\mathbb{1}_{\{\langle x,Uz\rangle>0\}}\mathrm{d}\mu(x)\ \mathrm{d}z\mathrm{d}\sigma(U).
\end{aligned} \tag{50}$$

Hence, using Proposition 7, we can write $(\tilde{R}\mu)(z, U) = (\mathcal{H}\tilde{\mu})(\tilde{U}z)$ where $\tilde{\mu} = J^T_\# O_\# \mu$.

Now, let $\mu \in \ker(\tilde{R})$, then for all $z \in S^1$, $U \in \mathbb{V}_{d,2}$, $\tilde{R}\mu(z, U) = \mathcal{H}\tilde{\mu}(\tilde{U}z) = 0$ and hence $\tilde{\mu} \in \ker(\mathcal{H}) = \{\tilde{\mu} \in \mathcal{M}_{\text{even}}(S^{d-2}), \ \tilde{\mu}(S^{d-2}) = 0\}$.

First, let's show that $\mu \in \mathcal{M}_{\text{even}}(S^{d-1})$. Let $f \in C(S^{d-1})$ and $U \in \mathbb{V}_{d,2}$, then, by using the same notation as in Propositions 6 and 7, we have

$$
\begin{aligned}
\langle \mu, f \rangle_{S^{d-1}} &= \int_{S^{d-1}} f(x)\mathrm{d}\mu(x) = \int_{S^{d-1}} \int_{S^1} f(x)\mathbb{1}_{\{z=P^U(x)\}} \, \mathrm{d}z \, \mathrm{d}\mu(x) \\
&= \int_{S^1} \int_{S^{d-1}} f(x)\mathbb{1}_{\{z=P^U(x)\}} \mathrm{d}\mu(x)\mathrm{d}z \\
&= \int_{S^1} \int_{F \cap S^{d-1}} f(x)\mathbb{1}_{\{\langle x, Uz \rangle > 0\}}\mathrm{d}\mu(x)\mathrm{d}z \quad \text{by Prop. 6} \\
&= \int_{S^1} \int_{S^{d-2}} \tilde{f}(y)\mathbb{1}_{\{\langle y, \tilde{U}z \rangle > 0\}}\mathrm{d}\tilde{\mu}(y)\mathrm{d}z \\
&= \int_{S^1} \langle \mathcal{H}\tilde{\mu}, \tilde{f} \rangle_{S^{d-2}} \, \mathrm{d}z \\
&= \int_{S^1} \langle \tilde{\mu}, \mathcal{H}\tilde{f} \rangle_{S^{d-2}} \, \mathrm{d}z \\
&= \int_{S^1} \langle \tilde{\mu}, (\mathcal{H}\tilde{f})^- \rangle_{S^{d-2}} \, \mathrm{d}z \quad \text{since } \tilde{\mu} \in \mathcal{M}_{\text{even}} \\
&= \int_{S^{d-1}} f^-(x)\mathrm{d}\mu(x) = \langle \mu, f^- \rangle_{S^{d-1}},
\end{aligned}
\tag{51}
$$

using for the last line all the opposite transformations. Therefore, $\mu \in \mathcal{M}_{\text{even}}(S^{d-1})$.

Now, we need to find on which set the measure is null. We have

$$
\begin{aligned}
&\forall z \in S^1, U \in \mathbb{V}_{d,2}, \ \tilde{\mu}(S^{d-2}) = 0 \\
&\iff \forall z \in S^1, U \in \mathbb{V}_{d,2}, \ \mu(O^{-1}((J^T)^{-1}(S^{d-2}))) = \mu(F \cap S^{d-1}) = 0.
\end{aligned}
\tag{52}
$$

Hence, we deduce that

$$
\begin{aligned}
\ker(\tilde{R}) = \{\mu \in \mathcal{M}_{\text{even}}(S^{d-1}), \ \forall U \in \mathbb{V}_{d,2}, \forall z \in S^1, F = \text{span}(UU^T)^\perp \cap \text{span}(Uz), \\
\mu(F \cap S^{d-1}) = 0\}.
\end{aligned}
\tag{53}
$$

Moreover, we have that $\cup_{U,z} F_{U,z} \cap S^{d-1} = \{H \cap S^{d-1} \subset \mathbb{R}^d, \ \dim(H) = d - 1\}$.

Indeed, on the one hand, let H an hyperplane, $x \in H \cap S^{d-1}$, $U \in \mathbb{V}_{d,2}$, and note $z = P^U(x)$. Then, $x \in F \cap S^{d-1}$ by Proposition 6 and $H \cap S^{d-1} \subset \cup_{U,z} F_{U,z}$.

On the other hand, let $U \in \mathbb{V}_{d,2}$, $z \in S^1$, $F$ is a hyperplane since $\dim(F) = d - 1$ and therefore $F \cap S^{d-1} \subset \{H, \ \dim(H) = d - 1\}$.

Finally, we deduce that

$$
\ker(\tilde{R}) = \big\{\mu \in \mathcal{M}_{\text{even}}(S^{d-1}), \ \forall H \in \mathcal{G}_{d,d-1}, \ \mu(H \cap S^{d-1})\big\}.
\tag{54}
$$

## A.8 PROOF OF PROPOSITION 5

Let $p \geq 1$. First, it is straightforward to see that for all $\mu, \nu \in \mathcal{P}_p(S^{d-1})$, $SSW_p(\mu, \nu) \geq 0$, $SSW_p(\mu, \nu) = SSW_p(\nu, \mu)$, $\mu = \nu \implies SSW_p(\mu, \nu) = 0$ and that we have the triangular

inequality since

$$
\begin{aligned}
\forall \mu, \nu, \alpha \in \mathcal{P}_p(S^{d-1}), \; SSW_p(\mu, \nu) &= \Big( \int_{\mathbb{V}_{d,2}} W_p^p(P_\#^U \mu, P_\#^U \nu) \, \mathrm{d}\sigma(U) \Big)^{\frac{1}{p}} \\
&\leq \Big( \int_{\mathbb{V}_{d,2}} \big( W_p(P_\#^U \mu, P_\#^U \alpha) + W_p(P_\#^U \alpha, P_\#^U \nu) \big)^p \, \mathrm{d}\sigma(U) \Big)^{\frac{1}{p}} \\
&\leq \Big( \int_{\mathbb{V}_{d,2}} W_p^p(P_\#^U \mu, P_\#^U \alpha) \, \mathrm{d}\sigma(U) \Big)^{\frac{1}{p}} \\
&\quad + \Big( \int_{\mathbb{V}_{d,2}} W_p^p(P_\#^U \alpha, P_\#^U \nu) \, \mathrm{d}\sigma(U) \Big)^{\frac{1}{p}} \\
&= SSW_p(\mu, \alpha) + SSW_p(\alpha, \nu),
\end{aligned}
\tag{55}
$$

using the triangular inequality for $W_p$ and the Minkowski inequality. Therefore, it is at least a pseudo-distance.

To be a distance, we also need $SSW_p(\mu, \nu) = 0 \implies \mu = \nu$. Suppose that $SSW_p(\mu, \nu) = 0$. Since, for all $U \in \mathbb{V}_{d,2}$, $W_p^p(P_\#^U \mu, P_\#^U \nu) \geq 0$, $SSW_p^p(\mu, \nu) = 0$ implies that for $\sigma$-ae $U \in \mathbb{V}_{d,2}$, $W_p^p(P_\#^U \mu, P_\#^U \nu) = 0$ and hence $P_\#^U \mu = P_\#^U \nu$ or $(\tilde{R}\mu)^U = (\tilde{R}\nu)^U$ for $\sigma$-ae $U \in \mathbb{V}_{d,2}$ since $W_p$ is a distance on the circle. Therefore, it is a distance on the sets of injectivity of $\tilde{R}$.

### A.9 ADDITIONAL PROPERTIES

In this Section, we derive additional properties of SSW. First, we will show that the weak convergence implies the convergence *w.r.t* SSW. Then, we will show that the sample complexity is independent of the dimension. Finally, we will derive the projection complexity of SSW.

**Convergence Properties.**

**Proposition 8.** *Let* $(\mu_k), \mu \in \mathcal{P}_p(S^{d-1})$ *such that* $\mu_k \xrightarrow[k\to\infty]{} \mu$*, then*

$$
SSW_p(\mu_k, \mu) \xrightarrow[k\to\infty]{} 0. \tag{56}
$$

*Proof.* Since the Wasserstein distance metrizes the weak convergence (Corollary 6.11 (Villani, 2009)), we have $P_\#^U \mu_k \xrightarrow[k\to\infty]{} P_\#^U \mu$ (by continuity) $\iff W_p^p(P_\#^U \mu_k, P_\#^U \mu) \xrightarrow[k\to\infty]{} 0$ and hence by the dominated convergence theorem, $SSW_p^p(\mu_k, \mu) \xrightarrow[k\to\infty]{} 0$. $\qquad \square$

**Sample Complexity.** We show here that the sample complexity is independent of the dimension. Actually, this is a well known properties of sliced-based distances and it was studied first in (Nadjahi et al., 2020). To the best of our knowledge, the sample complexity of the Wasserstein distance on the circle has not been previously derived. We suppose in the next proposition that it is known as we mainly want to show that the sample complexity of SSW does not depend on the dimension.

**Proposition 9.** *Let* $p \geq 1$*. Suppose that for* $\mu, \nu \in \mathcal{P}(S^1)$*, with empirical measures* $\hat{\mu}_n = \frac{1}{n} \sum_{i=1}^n \delta_{x_i}$ *and* $\hat{\nu}_n = \frac{1}{n} \sum_{i=1}^n \delta_{y_i}$*, where* $(x_i)_i \sim \mu$*,* $(y_i)_i \sim \nu$ *are independent samples, we have*

$$
\mathbb{E}[|W_p^p(\hat{\mu}_n, \hat{\nu}_n) - W_p^p(\mu, \nu)|] \leq \beta(p, n). \tag{57}
$$

*Then, for any* $\mu, \nu \in \mathcal{P}_{p,ac}(S^{d-1})$ *with empirical measures* $\hat{\mu}_n$ *and* $\hat{\nu}_n$*, we have*

$$
\mathbb{E}[|SSW_p^p(\hat{\mu}_n, \hat{\nu}_n) - SSW_p^p(\mu, \nu)|] \leq \beta(p, n). \tag{58}
$$

*Proof.* By using the triangle inequality, Fubini-Tonelli, and the hypothesis on the sample complexity of $W_p^p$ on $S^1$, we obtain:

$$
\begin{aligned}
\mathbb{E}[|SSW_p^p(\hat{\mu}_n, \hat{\nu}_n) - SSW_p^p(\mu, \nu)|] &= \mathbb{E}\left[\left|\int_{\mathbb{V}_{d,2}} \left(W_p^p(P_\#^U \hat{\mu}_n, P_\#^U \hat{\nu}_n) - W_p^p(P_\#^U \mu, P_\#^U \nu)\right) \mathrm{d}\sigma(U)\right|\right] \\
&\leq \mathbb{E}\left[\int_{\mathbb{V}_{d,2}} \left|W_p^p(P_\#^U \hat{\mu}_n, P_\#^U \hat{\nu}_n) - W_p^p(P_\#^U \mu, P_\#^U \nu)\right| \mathrm{d}\sigma(U)\right] \\
&= \int_{\mathbb{V}_{d,2}} \mathbb{E}\left[\left|W_p^p(P_\#^U \hat{\mu}_n, P_\#^U \hat{\nu}_n) - W_p^p(P_\#^U \mu, P_\#^U \nu)\right|\right] \mathrm{d}\sigma(U) \\
&\leq \int_{\mathbb{V}_{d,2}} \beta(p, n) \, \mathrm{d}\sigma(U) \\
&= \beta(p, n).
\end{aligned}
\tag{59}
$$

$\square$

**Projection Complexity.** We derive in the next proposition the projection complexity, which refers to the convergence rate of the Monte Carlo approximate *w.r.t* of the number of projections $L$ towards the true integral. Note that we find the typical rate of Monte Carlo estimates, and that it has already been derive for sliced-based distances in (Nadjahi et al., 2020).

**Proposition 10.** *Let $p \geq 1$, $\mu, \nu \in \mathcal{P}_{p,ac}(S^{d-1})$. Then, the error made with the Monte Carlo estimate of $SSW_p$ can be bounded as*

$$
\begin{aligned}
\mathbb{E}_U\left[\left|\widehat{SSW}_{p,L}^p(\mu, \nu) - SSW_p^p(\mu, \nu)\right|\right]^2 &\leq \frac{1}{L}\int_{\mathbb{V}_{d,2}} \left(W_p^p(P_\#^U \mu, P_\#^U \nu) - SSW_p^p(\mu, \nu)\right)^2 \mathrm{d}\sigma(U) \\
&= \frac{1}{L}\mathrm{Var}_U\left(W_p^p(P_\#^U \mu, P_\#^U \nu)\right),
\end{aligned}
\tag{60}
$$

*where $\widehat{SSW}_{p,L}^p(\mu, \nu) = \frac{1}{L}\sum_{i=1}^L W_p^p(P_\#^{U_i} \mu, P_\#^{U_i} \nu)$ with $(U_i)_{i=1}^L \sim \sigma$ independent samples.*

*Proof.* Let $(U_i)_{i=1}^L$ be iid samples of $\sigma$. Then, by first using Jensen inequality and then remembering that $\mathbb{E}_U[W_p^p(P_\#^U \mu, P_\#^U \nu)] = SSW_p^p(\mu, \nu)$, we have

$$
\begin{aligned}
\mathbb{E}_U\left[\left|\widehat{SSW}_{p,L}^p(\mu, \nu) - SSW_p^p(\mu, \nu)\right|\right]^2 &\leq \mathbb{E}_U\left[\left|\widehat{SSW}_{p,L}^p(\mu, \nu) - SSW_p^p(\mu, \nu)\right|^2\right] \\
&= \mathbb{E}_U\left[\left|\frac{1}{L}\sum_{i=1}^L \left(W_p^p(P_\#^{U_i} \mu, P_\#^{U_i} \nu) - SSW_p^p(\mu, \nu)\right)\right|^2\right] \\
&= \frac{1}{L^2}\mathrm{Var}_U\left(\sum_{i=1}^L W_p^p(P_\#^{U_i} \mu, P_\#^{U_i} \nu)\right) \\
&= \frac{1}{L}\mathrm{Var}_U\left(W_p^p(P_\#^U \mu, P_\#^U \nu)\right) \\
&= \frac{1}{L}\int_{\mathbb{V}_{d,2}} \left(W_p^p(P_\#^U \mu, P_\#^U \nu) - SSW_p^p(\mu, \nu)\right)^2 \mathrm{d}\sigma(U).
\end{aligned}
\tag{61}
$$

$\square$

# B  BACKGROUND ON THE SPHERE

## B.1  UNIQUENESS OF THE PROJECTION

Here, we discuss the uniqueness of the projection $P^U$ for almost every $x$. For that, we recall some results of (Bardelli & Mennucci, 2017).

Let $M$ be a closed subset of a complete finite-dimensional Riemannian manifold $N$. Let $d$ be the Riemannian distance on $N$. Then, the distance from the set $M$ is defined as

$$d_M(x) = \inf_{y \in M} d(x, y). \tag{62}$$

The infimum is a minimum since $M$ is closed and $N$ locally compact, but the minimum might not be unique. When it is unique, let's denote the point which attains the minimum as $\pi(x)$, *i.e.* $d(x, \pi(x)) = d_M(x)$.

**Proposition 11** (Proposition 4.2 in (Bardelli & Mennucci, 2017)). *Let $M$ be a closed set in a complete $m$-dimensional Riemannian manifold $N$. Then, for almost every $x$, there exists a unique point $\pi(x) \in M$ that realizes the minimum of the distance from $x$.*

From this Proposition, they further deduce that the measure $\pi_\# \gamma$ is well defined on $M$ with $\gamma$ a locally absolutely continuous measure *w.r.t.* the Lebesgue measure.

In our setting, for all $U \in \mathbb{V}_{d,2}$, we want to project a measure $\mu \in \mathcal{P}(S^{d-1})$ on the great circle $\text{span}(UU^T) \cap S^{-1}$. Hence, we have $N = S^{d-1}$ which is a complete finite-dimensional Riemannian manifold and $M = \text{span}(UU^T) \cap S^{d-1}$ a closed set in $N$. Therefore, we can apply Proposition 11 and the push-forward measures are well defined for absolutely continuous measures.

## B.2 Optimization on the Sphere

Let $F : S^{d-1} \to \mathbb{R}$ be some functional on the sphere. Then, we can perform a gradient descent on a Riemannian manifold by following the geodesics, which are the counterpart of straight lines in $\mathbb{R}^d$. Hence, the gradient descent algorithm (Absil et al., 2009; Bonnabel, 2013) reads as

$$\forall k \geq 0, \ x_{k+1} = \exp_{x_k}\left(-\gamma \text{grad} f(x)\right), \tag{63}$$

where for all $x \in S^{d-1}$, $\exp_x : T_x S^{d-1} \to S^{d-1}$ is a map from the tangent space $T_x S^{d-1} = \{v \in \mathbb{R}^d, \ \langle x, v \rangle = 0\}$ to $S^{d-1}$ such that for all $v \in T_x S^{d-1}$, $\exp_x(v) = \gamma_v(1)$ with $\gamma_v$ the unique geodesic starting from $x$ with speed $v$, *i.e.* $\gamma(0) = x$ and $\gamma'(0) = v$.

For $S^{d-1}$, the exponential map is known and is

$$\forall x \in S^{d-1}, \forall v \in T_x S^{d-1}, \ \exp_x(v) = \cos(\|v\|_2)x + \sin(\|v\|_2)\frac{v}{\|v\|_2}. \tag{64}$$

Moreover, the Riemannian gradient on $S^{d-1}$ is known as (Absil et al., 2009, Eq. 3.37)

$$\text{grad} f(x) = \text{Proj}_x(\nabla f(x)) = \nabla f(x) - \langle \nabla f(x), x \rangle x, \tag{65}$$

$\text{Proj}_x$ denoting the orthogonal projection on $T_x S^{d-1}$.

For more details, we refer to (Absil et al., 2009; Boumal, 2022).

## B.3 Von Mises-Fisher Distribution

The von Mises-Fisher (vMF) distribution is a distribution on $S^{d-1}$ characterized by a concentration parameter $\kappa > 0$ and a location parameter $\mu \in S^{d-1}$ through the density

$$\forall \theta \in S^{d-1}, \ f_{\text{vMF}}(\theta; \mu, \kappa) = \frac{\kappa^{d/2-1}}{(2\pi)^{d/2} I_{d/2-1}(\kappa)} \exp(\kappa \mu^T \theta), \tag{66}$$

where $I_\nu(\kappa) = \frac{1}{2\pi} \int_0^\pi \exp(\kappa \cos(\theta)) \cos(\nu \theta) d\theta$ is the modified Bessel function of the first kind.

Several algorithms allow to sample from it, see *e.g.* (Wood, 1994; Ulrich, 1984) for algorithms using rejection sampling or (Kurz & Hanebeck, 2015) without rejection sampling.

For $d = 1$, the vMF coincides with the von Mises (vM) distribution, which has for density

$$\forall \theta \in [-\pi, \pi[, \ f_{\text{vM}}(\theta; \mu, \kappa) = \frac{1}{I_0(\kappa)} \exp(\kappa \cos(\theta - \mu)), \tag{67}$$

with $\mu \in [0, 2\pi[$ the mean direction and $\kappa > 0$ its concentration parameter. We refer to (Mardia et al., 2000, Section 3.5 and Chapter 9) for more details on these distributions.

In particular, for $\kappa = 0$, the vMF (resp. vM) distribution coincides with the uniform distribution on the sphere (resp. the circle).

Jung (2021) studied the law of the projection of a vMF on a great circle. In particular, they showed that, while the vMF plays the role of the normal distributions for directional data, the projection actually does not follow a von Mises distribution. More precisely, they showed the following theorem:

**Theorem 1** (Theorem 3.1 in (Jung, 2021)). *Let $d \geq 3$, $X \sim \text{vMF}(\mu, \kappa) \in S^{d-1}$, $U \in \mathbb{V}_{d,2}$ and $T = P^U(X)$ the projection on the great circle generated by $U$. Then, the density function of $T$ is*

$$\forall t \in [-\pi, \pi[, \ f(t) = \int_0^1 f_R(r) f_{\text{vM}}(t; 0, \kappa \cos(\delta)r) \ \mathrm{d}r, \tag{68}$$

*where $\delta$ is the deviation of the great circle (geodesic) from $\mu$ and the mixing density is*

$$\forall r \in ]0, 1[, \ f_R(r) = \frac{2}{I_\nu^*(\kappa)} I_0(\kappa \cos(\delta)r) r (1 - r^2)^{\nu-1} I_{\nu-1}^*(\kappa \sin(\delta)\sqrt{1-r^2}), \tag{69}$$

*with $\nu = (d-2)/2$ and $I_\nu^*(z) = (\frac{z}{2})^{-\nu} I_\nu(z)$ for $z > 0$, $I_\nu^*(0) = 1/\Gamma(\nu+1)$.*

Hence, as noticed by Jung (2021), in the particular case $\kappa = 0$, *i.e.* $X \sim \text{Unif}(S^{d-1})$, then

$$f(t) = \int_0^1 f_R(r) f_{\text{vM}}(t; 0, 0) \ \mathrm{d}r = f_{\text{vM}(t;0,0)} \int_0^1 f_R(r) \mathrm{d}r = f_{\text{vM}}(t; 0, 0), \tag{70}$$

and hence $T \sim \text{Unif}(S^1)$.

## B.4 Normalizing Flows on the Sphere

Normalizing flows (Papamakarios et al., 2021) are invertible transformations. There has been a recent interest in defining such transformations on manifolds, and in particular on the sphere (Rezende et al., 2020; Cohen et al., 2021; Rezende & Racanière, 2021).

**Exponential map normalizing flows.** Here, we implemented the Exponential map normalizing flows introduced in (Rezende et al., 2020). The transformation $T$ is

$$\forall x \in S^{d-1}, \ z = T(x) = \exp_x\left(\text{Proj}_x(\nabla\phi(x))\right), \tag{71}$$

where $\phi(x) = \sum_{i=1}^K \frac{\alpha_i}{\beta_i} e^{\beta_i(x^T \mu_i - 1)}$, $\alpha_i \geq 0$, $\sum_i \alpha_i \leq 1$, $\mu_i \in S^{d-1}$ and $\beta_i > 0$ for all $i$. $(\alpha_i)_i$, $(\beta_i)_i$ and $(\mu_i)_i$ are the learnable parameters.

The density of $z$ can be obtained as

$$p_Z(z) = p_X(x) \det\left(E(x)^T J_T(x)^T J_T(x) E(x)\right)^{-\frac{1}{2}}, \tag{72}$$

where $J_f$ is the Jacobian in the embedded space and $E(x)$ it the matrix whose columns form an orthonormal basis of $T_x S^{d-1}$.

The common way of training normalizing flows is to use either the reverse or forward KL divergence. Here, we use them with a different loss, namely SSW.

**Stereographic projection.** The stereographic projection $\rho : S^{d-1} \to \mathbb{R}^{d-1}$ maps the sphere $S^{d-1}$ to the Euclidean space. A strategy first introduced in (Gemici et al., 2016) is to use it before applying a normalizing flows in the Euclidean space in order to map some prior, and which allows to perform density estimation.

More precisely, the stereographic projection is defined as

$$\forall x \in S^{d-1}, \ \rho(x) = \frac{x_{2:d}}{1 + x_1}, \tag{73}$$

and its inverse is

$$\forall u \in \mathbb{R}^{d-1}, \ \rho^{-1}(u) = \begin{pmatrix} 2\frac{u}{\|u\|_2^2 + 1} \\ 1 - \frac{2}{\|u\|_2^2 + 1} \end{pmatrix}. \tag{74}$$

Gemici et al. (2016) derived the change of variable formula for this transformation, which comes from the theory of probability between manifolds. If we have a transformation $T = f \circ \rho$, where $f$ is a normalizing flows on $\mathbb{R}^{d-1}$, *e.g.* a RealNVP (Dinh et al., 2016), then the log density of the target distribution can be obtained as

$$
\begin{aligned}
\log p(x) &= \log p_Z(z) + \log |\det J_f(z)| - \frac{1}{2}\log |\det J_{\rho^{-1}}^T J_{\rho^{-1}}(\rho(x))| \\
&= \log p_Z(z) + \log |\det J_f(z)| - d \log \left( \frac{2}{\|\rho(x)\|_2^2 + 1} \right),
\end{aligned}
\tag{75}
$$

where we used the formula of (Gemici et al., 2016) for the change of variable formula of $\rho$, and where $p_Z$ is the density of some prior on $\mathbb{R}^{d-1}$, typically of a standard Gaussian. We refer to (Gemici et al., 2016; Mathieu & Nickel, 2020) for more details about these transformations.

## C  ADDITIONAL EXPERIMENTS

### C.1  EVOLUTION OF SSW BETWEEN VON MISES-FISHER DISTRIBUTIONS

The KL divergence between the von Mises-Fisher distribution and the uniform distribution has been derived analytically in (Davidson et al., 2018; Xu & Durrett, 2018) as

$$
\begin{aligned}
\mathrm{KL}\big(\mathrm{vMF}(\mu, \kappa)||\mathrm{vMF}(\cdot, 0)\big) &= \kappa \frac{I_{d/2}(\kappa)}{I_{d/2-1}(\kappa)} + \left( \frac{d}{2} - 1 \right) \log \kappa - \frac{d}{2}\log(2\pi) - \log I_{d/2-1}(\kappa) \\
&\quad + \frac{d}{2}\log \pi + \log 2 - \log \Gamma\left( \frac{d}{2} \right).
\end{aligned}
\tag{76}
$$

We plot on Figure 7 the evolution of KL and SSW *w.r.t.* $\kappa$ for different dimensions. We observe a different trend. SSW seems to get lower with the dimension contrary to KL.

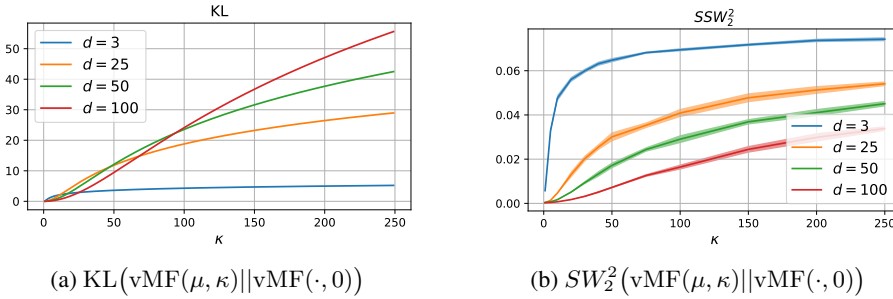

(a) $\mathrm{KL}\big(\mathrm{vMF}(\mu, \kappa)||\mathrm{vMF}(\cdot, 0)\big)$  (b) $SW_2^2\big(\mathrm{vMF}(\mu, \kappa)||\mathrm{vMF}(\cdot, 0)\big)$

Figure 7: Evolution *w.r.t* $\kappa$ between $\mathrm{vMK}(\mu, \kappa)$ and $\mathrm{vMF}(\cdot, 0)$. For SW, we used 100 projections (for memory reasons for $d = 100$), and computed it for $\kappa \in \{1, 5, 10, 20, 30, 40, 50, 75, 100, 150, 200, 250\}$, 10 times by dimension and $\kappa$, and with 500 samples of both distributions.

As a sanity check, we compare on Figure 8 the evolution of SSW between vMF distributions where we fix $\mathrm{vMF}(\mu_0, 10)$ and we rotate the first vMF along a great circle. More precisely, we plot $SW_2^2\big(\mathrm{vMF}((1, 0, 0, ...), 10), \mathrm{vMF}((\cos(\theta), \sin(\theta), 0, ...), 10)\big)$ for $\theta \in \{\frac{k\pi}{6}\}_{k \in \{0, ..., 12\}}$. As expected, we obtain a bell shape which is maximal when the second vMF distribution has for location parameter $-\mu_0$. We observe a similar behavior between $SSW_2$, $SSW_1$ and $SW_2$ with different scales.

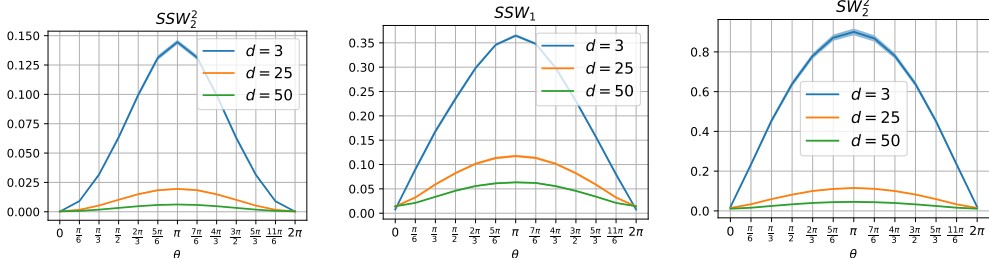

Figure 8: Evolution of $SW$ between vMF samples in $S^{d-1}$ (mean over 100 batch).

On Figure 9, we plot the evolution of SSW *w.r.t.* the number of projections for different dimensions. We observe that for around 100 projections, the variance seems to be low enough.

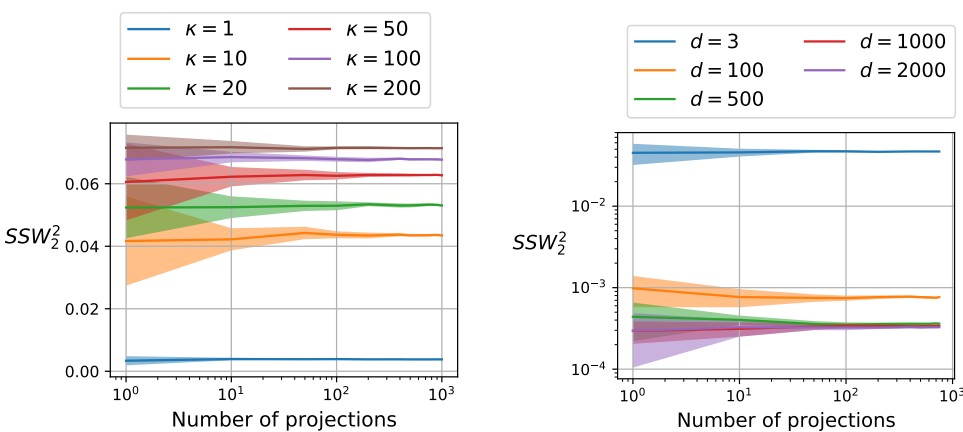

Figure 9: Influence of the number of projections. We compute $SW_2^2\big(\text{vMF}(\mu,\kappa)||\text{vMF}(\cdot,0)\big)$ 20 times, for $n = 500$ samples in dimension $d = 3$.

Nadjahi et al. (2020) proved that, contrary to the Wasserstein distance, the classical sliced-Wasserstein distance has a sample complexity independent of the dimension $d$. As shown in Propositon 9, we have similar results for SSW. We show it empirically on Figure 10 by plotting SSW and the Wasserstein distance (with geodesic distance) between samples of the uniform distribution on the sphere *w.r.t.* the number of samples. We observe indeed that the convergence rate of SSW is independent of the dimension.

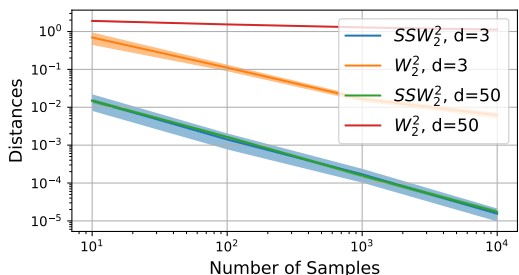

Figure 10: Spherical Sliced-Wasserstein and Wasserstein distance (with geodesic distance) between samples of the uniform distribution on the sphere. Results are averaged over 20 runs and the shaded are correponds to the standard deviation.

## C.2   RUNTIME COMPARISONS

We study here the evolution of the runtime *w.r.t.* different parameters. On Figure 11, we plot for several dimensions the runtime to compute $SSW_2$ *w.r.t.* the number of projections and the number of samples. We observe the linearity *w.r.t.* the number of projections and the quasi-linearity *w.r.t.* the number of samples.

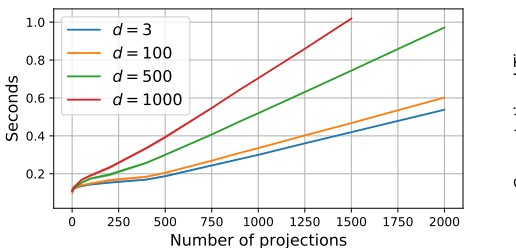 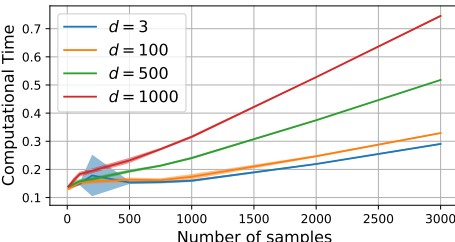

Figure 11: Computation time *w.r.t.* the number of projections or samples, taken for $\kappa = 10$ and $n = 500$ samples for the left figure, and $\kappa = 10$ and 200 projections for the right figure, and for 20 times.

## C.3   GRADIENT FLOWS

**Mixture of vMF distributions.**   For the experiment in Section 5.1, we use as target distribution of mixture of 6 vMF distributions from which we have access to samples. We refer to Appendix B.3 for background on vMF distributions.

The 6 vMF distributions have weights $1/6$, concentration parameter $\kappa = 10$ and location parameters $\mu_1 = (1, 0, 0)$, $\mu_2 = (0, 1, 0)$, $\mu_3 = (0, 0, 1)$, $\mu_4 = (-1, 0, 0)$, $\mu_5 = (0, -1, 0)$ and $\mu_6 = (0, 0, -1)$.

We use two different approximation of the distribution. First, we approximate it using the empirical distribution, *i.e.* $\hat{\mu} = \frac{1}{n} \sum_{i=1}^{n} \delta_{x_i}$ and we optimize over the particles $(x_i)_{i=1}^{n}$. To optimize over particles, we can either use a projected gradient descent:

$$\begin{cases} x^{(k+1)} = x^{(k)} - \gamma \nabla_{x^{(k)}} SSW_2^2(\hat{\mu}_k, \nu) \\ x^{(k+1)} = \frac{x^{(k+1)}}{\|x^{(k+1)}\|_2}, \end{cases} \tag{77}$$

or a Riemannian gradient descent on the sphere (Absil et al., 2009) (see Appendix B.2 for more details). Note that the projected gradient descent is a Riemannian gradient descent with retraction (Boumal, 2022).

We can also use neural networks such as a multilayer perceptron (MLP). We used a MLP composed of 5 layers of 100 units with leaky relu activation functions. The output of the MLP is normalized on the sphere using a $\ell^2$ normalization. We perform a gradient descent using Adam (Kingma & Ba, 2014) as the optimizer with a learning rate of $10^{-4}$ for 2000 epochs. We approximate SSW with $L = 1000$ projections and a batch size of 500. The base distribution is choose as the uniform distribution on the sphere.

We report on Figure 12 a comparison of the 2 approximations where the density is estimated with a Gaussian kernel density estimator.

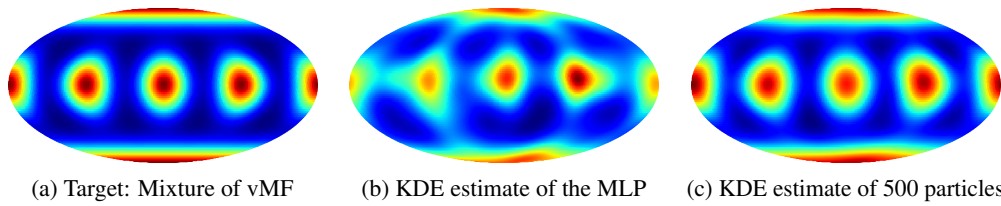

(a) Target: Mixture of vMF          (b) KDE estimate of the MLP          (c) KDE estimate of 500 particles

Figure 12: Minimization of SSW with respect to a mixture of vMF.

**vMF distribution.** A a simpler experiment, we choose a simple vMF distribution with $\kappa = 10$. We report on Figure 13 the evolution of the density approximated using a KDE, and on Figure 14 the evolution of particles.

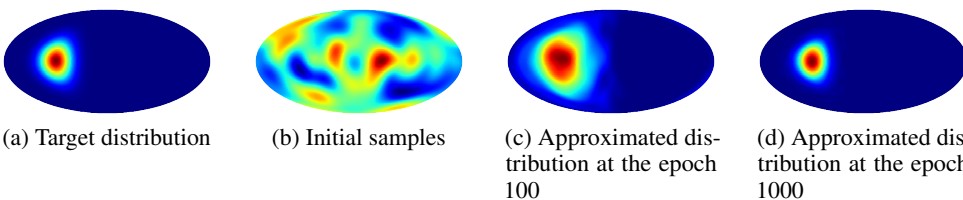

(a) Target distribution    (b) Initial samples    (c) Approximated distribution at the epoch 100    (d) Approximated distribution at the epoch 1000

Figure 13: Gradient Flows on SW with a vMF target and Mollweide projections. The distributions are approximated using KDE.

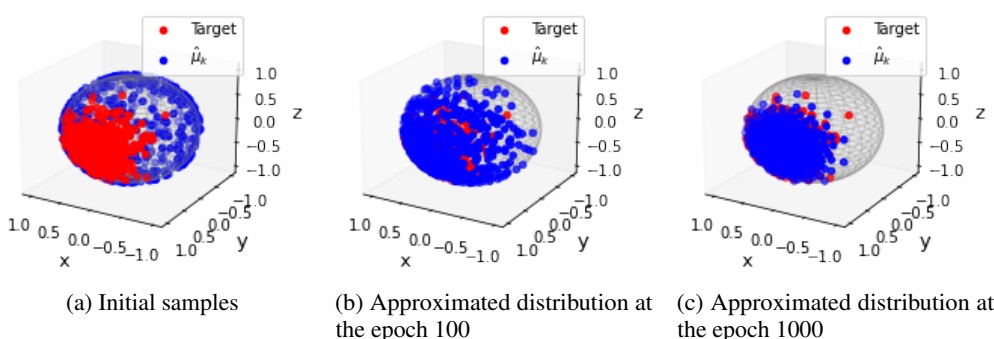

(a) Initial samples    (b) Approximated distribution at the epoch 100    (c) Approximated distribution at the epoch 1000

Figure 14: Gradient Flows on SW with a vMF target and Mollweide projections.

## C.4    EARTH DATA ESTIMATION

Let $T$ be a normalizing flow (NF). For a density estimation task, we have access to a distribution $\mu$ through samples $(x_i)_{i=1}^n$, *i.e.* through the empirical measure $\hat{\mu}_n = \frac{1}{n}\sum_{i=1}^n \delta_{x_i}$ . And the goal is to find an invertible transformation $T$ such that $T_{\#}\mu = p_Z$, where $p_Z$ is a prior distribution for which we know the density. In that case, indeed, the density of $\mu$, denoted as $f_\mu$ can be obtained as

$$\forall x, \ f_\mu(x) = p_Z(T(x))|\det J_T(x)|. \tag{78}$$

For the invertible transform, we propose to use normalizing flows on the sphere (see Appendix B.4). We use two different normalizing flows, exponential map normalizing flows (Rezende et al., 2020) and Real NVP (Dinh et al., 2016) + stereographic projection (Gemici et al., 2016) which we call "Stereo" in Table 1.

To fit $T_{\#}\mu = p_Z$, we use either SSW, SW on the sphere, or SW on $\mathbb{R}^{d-1}$ for the stereographic projection based NF. For the exponential map normalizing flow, we compose 48 blocks, each one with 100 components. These transformations have 24000 parameters. For Real NVP, we compose 10 blocks of Real NVPs, with shifting and scaling as multilayer perceptron, composed of 10 layers, 25 hidden units and with leaky relu of parameters 0.2 for the activation function. The number of parameters of these networks are 27520.

For the training process, we perform 20000 epochs with full batch size. We use Adam as optimizer with a learning rate of $10^{-1}$. For the sterographic NF, we use a learning rate of $10^{-3}$.

We report in Table 3 details of the datasets.

---

**Algorithm 2** SWVI (Yi & Liu, 2021)

---

**Input:** $V$ a potential, $K$ the number of iterations of SWVI, $N$ the batch size, $\ell$ the number of MCMC steps
**Initialization:** Choose $q_\theta$ a sampler
**for** $k = 1$ **to** $K$ **do**
    Sample $(z_i^0)_{i=1}^N \sim q_\theta$
    Run $\ell$ MCMC steps starting from $(z_i^0)_{i=1}^N$ to get $(z_j^\ell)_{j=1}^N$
    // Denote $\hat{\mu}_0 = \frac{1}{N} \sum_{j=1}^N \delta_{z_j^0}$ and $\hat{\mu}_\ell = \frac{1}{N} \sum_{j=1}^N \delta_{z_j^\ell}$
    Compute $J = SW_2^2(\hat{\mu}_0, \hat{\mu}_\ell)$
    Backpropagate through $J$ *w.r.t.* $\theta$
    Perform a gradient step
**end for**

---

Table 3: Details of Earth datasets.

|  | Earthquake | Flood | Fire |
|---|---|---|---|
| Train set size | 4284 | 3412 | 8966 |
| Test set size | 1836 | 1463 | 3843 |
| Data size | 6120 | 4875 | 12809 |

## C.5 SLICED-WASSERSTEIN VARIATIONAL INFERENCE

### C.5.1 VARIATIONAL INFERENCE

In variational inference (VI) (Jordan et al., 1999; Blei et al., 2017), we have some observed data $(x_i)_{i=1}^n$ and some latent data $(z_i)_{i=1}^n$. The goal of variational inference is to approximate the posterior distribution $p(\cdot|x)$ by some distribution $q \in \mathcal{Q}$ where $\mathcal{Q}$ is a family of probabilities. The usual way of doing that is to minimize the Kullback-Leibler divergence among this family, *i.e.*

$$\min_{q \in \mathcal{Q}} \mathrm{KL}(q || p(\cdot|x)) = \mathbb{E}_q[\log\left(\frac{q(Z)}{p(Z|x)}\right)]. \tag{79}$$

But the KL divergence suffers from some drawbacks, as it is only a divergence (*i.e.* it does not satisfy the triangular inequality, and it is non symmetric), but it also suffers from under estimating the target distribution (or over estimating it for the reverse KL).

Yi & Liu (2021) propose to use an optimal transport distance instead, namely the SW distance which gives the sliced-Wasserstein variational inference method. Basically, given some unnormalized probability $p(\cdot|x)$ that we want to approximate with some variational distribution $q_\phi$, we can first apply a MCMC algorithm and then learn $q_\phi$ using a gradient descent on SW with the target being the empirical distributions of the samples given by the MCMC. But running long MCMC chain is time consuming and it might be difficult to diagnose burn-in period. Therefore, they propose to only run at each iteration some number of steps $t$ of MCMC chain, and then learn by gradient descent the variational distribution. Therefore, the variational distribution is guided at each step by the MCMC samples toward the stationary distribution which is the target. This is called an amortized sampler (see Problem 1 in (Wang & Liu, 2016)). We sum up the procedure in Algorithm 2.

We propose here to substitute $SW$ by $SSW$ in order to perform SSWVI on the sphere. To do that, we first need a MCMC method on the sphere.

### C.5.2 MCMC ON THE SPHERE

Several MCMC methods on the sphere have been proposed. For example, Hamiltonian Monte-Carlo (HMC) methods were proposed in (Byrne & Girolami, 2013; Lan et al., 2014; Liu et al., 2016), and Riemannian Langevin algorithms were proposed in (Li & Erdogdu, 2020; Wang et al., 2020).

In our experiments, we use the Geodesic Langevin algorithm (GLA) introduced by Wang et al. (2020). This algorithm is a natural generalization of the Unadjusted Langevin Algorithm (ULA) and

it consists at simply following the geodesics of the regular ULA step, *i.e.*

$$\forall k > 0, \; x_{k+1} = \exp_{x_k}\left(\mathrm{Proj}_{x_k}(-\gamma \nabla V(x_k) + \sqrt{2\gamma}Z)\right), \; Z \sim \mathcal{N}(0, I), \tag{80}$$

where for the sphere,

$$\forall x \in S^{d-1}, \forall v \in T_x S^{d-1}, \; \exp_x(v) = x \cos(\|v\|) + \frac{v}{\|v\|} \sin(\|v\|), \tag{81}$$

$\mathrm{Proj}_x$ is the projection on the tangent space $T_x S^{d-1} = \{v \in \mathbb{R}^d, \; \langle x, v \rangle = 0\}$ (which is the orthogonal space) and is defined as

$$\mathrm{Proj}_x(v) = v - \langle x, v \rangle x. \tag{82}$$

For more details, we refer to (Absil et al., 2009).

We use GLA here for simplicity and as a proof of concept. But note that GLA, as ULA, is biased and therefore the distribution learned will not be the exact true stationary distribution. However, a Metropolis-Hastings step at each iteration could be used to enforce the reversibility *w.r.t.* the target distribution or we could use other MCMC with more appealing convergence properties (see *e.g.* Liu et al. (2016)).

### C.5.3 APPLICATIONS

**Target: Power spherical distribution.**    First, as a simple example on $S^2$, we use the power spherical distribution introduced by De Cao & Aziz (2020). This distribution has the advantage over the vMF distribution to allow for the direct use of the reparameterization trick since it does not require rejection sampling. The pdf is obtained as,

$$\forall x \in S^{d-1}, \; p_X(x; \mu, \kappa) \propto (1 + \mu^T x)^\kappa \tag{83}$$

with $\mu \in S^{d-1}$ and $\kappa > 0$. We can sample from drawing first $Z \sim \mathrm{Beta}(\frac{d-1}{2} + \kappa, \frac{d-1}{2})$, $v \sim \mathrm{Unif}(S^{d-2})$, then constructing $T = 2Z - 1$ and $Y = [T, v^T \sqrt{1 - T^2}]^T$. Finally, apply a Householder reflection about $\mu$ to $Y$. All the operations are well differentiable and allow to apply the reparametrization trick. For the algorithm, see Algorithm 1 in (De Cao & Aziz, 2020). Hence, in this case, if we denote $g_\theta$ the map which takes samples from a uniform distribution on $S^{d-2}$ and from a Beta distribution as input and outputs samples of power spherical distribution with parameters $\theta = (\kappa, \mu)$, we can use it as the sampler. We test the algorithm with a target being a power spherical distribution of parameter $\mu = (0, 1, 0)$ and $\kappa = 10$, starting from $\mu = (1, 1, 1)$ and $\kappa = 0.1$. Performing 2000 optimization steps with a gradient descent (Riemannian gradient descent on $\mu$ to stay on the sphere), and 20 steps of the GLA algorithm, we are getting close enough to the true distribution as we can see on Figure 15.

For the hyperparameters, we used a step size of $10^{-3}$ for GLA, 1000 projections to approximate SSW, a Riemannian gradient descent on the sphere (Absil et al., 2009) to learn the location parameter $\mu$ with a learning rate of 2, and a learning of 200 for $\kappa$. We performed $K = 2000$ steps and used $N = 500$ particles.

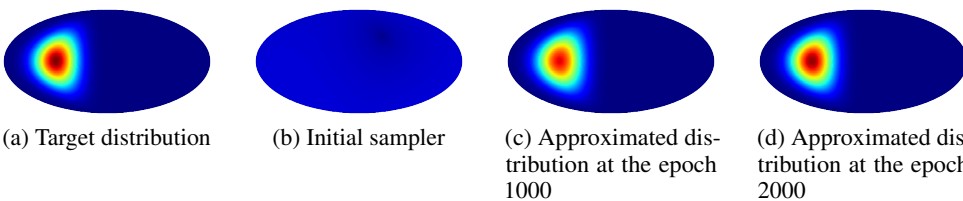

(a) Target distribution        (b) Initial sampler        (c) Approximated distribution at the epoch 1000        (d) Approximated distribution at the epoch 2000

Figure 15: SWVI on Power Spherical Distributions with Mollweide projections.

**Target: mixture of vMFs.**    In Section 5.1, we perform amortized variational inference with a mixture of vMF distributions as target. For this, we train exponential map normalizing flows (see (Rezende et al., 2020) and Appendix B.4). Moreover, we use the same target as Rezende et al. (2020),

*i.e.* the target $\nu$ has a density $p(x) \propto \sum_{k=1}^{4} e^{10x^T T_{s \to e}(\mu_k)}$ with $\mu_1 = (0.7, 1.5)$, $\mu_2 = (-1, 1)$, $\mu_3 = (0.6, 0.5)$ and $\mu_4 = (-0.7, 4)$. These are spherical coordinates which are be converted to euclidean using $T_{s \to e}(\theta, \phi) = (\sin \phi \cos \theta, \sin \phi \sin \theta, \cos \phi)$.

The exponential map normalizing flow is composed of $N = 6$ blocks with $K = 5$ components. We run the algorithm for 10000 iterations, with at each iteration 20 steps of GLA with $\gamma = 10^{-1}$ as learning rate, and one step of backpropagation through SSW using the Adam (Kingma & Ba, 2014) optimizer with a learning rate of $10^{-3}$.

We report on Figure 16 the Mollweide projection of the learned density. Since we learn to samples from a noise distribution, here the uniform distribution on the sphere, we do not have directly access to the density and we report a kernel density estimate with a Gaussian kernel using the implementation of Scipy (Virtanen et al., 2020).

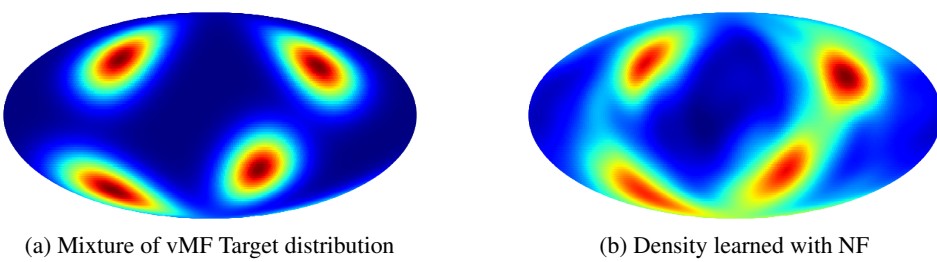

(a) Mixture of vMF Target distribution       (b) Density learned with NF

Figure 16: SSWVI on mixture of vMF

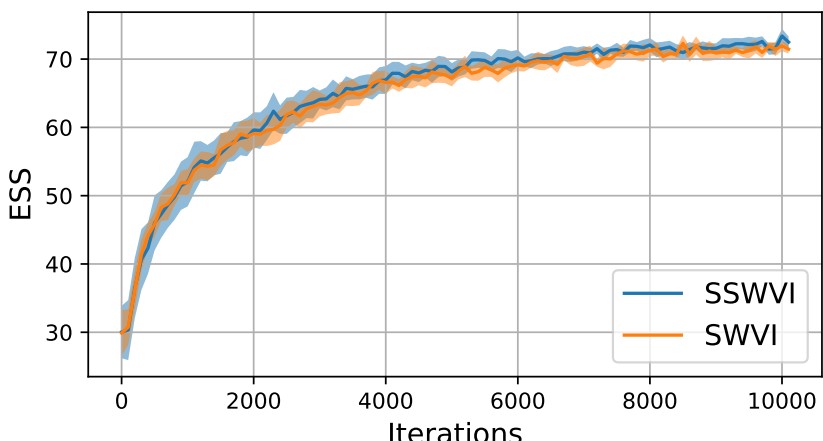

Figure 17: Comparison of the ESS between SWVI et SSWVI with the mixture target (mean over 10 runs).

We also report in Figure 17 the effective sample size (ESS) (Doucet et al., 2001; Liu & Chen, 1995) over the iterations. The ESS is estimated by (Rezende et al., 2020)

$$\text{ESS} = \frac{\text{Var}_{Unif}\left(e^{-\beta u(X)}\right)}{\text{Var}_q\left(\frac{e^{-\beta u(X)}}{q_\eta(X)}\right)} \approx \frac{\left(\sum_{s=1}^{S} w_s\right)^2}{\sum_{s=1}^{S} w_s^2}, \tag{84}$$

where $w_s = e^{-\beta u(x_s)}/q_\eta(x_s)$. The ESS is reported as a percentage of the sample size. Higher ESS indicates that the flow matches the target better (Rezende et al., 2020).

### C.6    SLICED-WASSERSTEIN AUTOENCODER

We recall that in the WAE framework, we want to minimize

$$\mathcal{L}(f,g) = \int c\big(x, g(f(x))\big)\mathrm{d}\mu(x) + \lambda D(f_\# \mu, p_Z), \tag{85}$$

where $f$ is an encoder, $g$ a decoder, $p_Z$ a prior distribution, $c$ some cost function and $D$ is a divergence in the latent space. Several $D$ were proposed. For example, Tolstikhin et al. (2018) proposed to use the MMD, Kolouri et al. (2018) used the SW distance, Patrini et al. (2020) used the Sinkhorn divergence, Kolouri et al. (2019) used the generalized SW distance. Here, we use $D = SSW_2^2$.

**Architecture and procedure.**    We first detail the hyperparameters and architectures of neural networks for MNIST and Fashion MNIST. For the encoder $f$ and the decoder $g$, we use the same architecture as Kolouri et al. (2018).

For both the encoder and the decoder architecture, we use fully convolutional architectures with 3x3 convolutional filters. More precisely, the architecture of the encoder is

$$
\begin{aligned}
x \in \mathbb{R}^{28\times28} &\to \mathrm{Conv2d}_{16} \to \mathrm{LeakyReLU}_{0.2} \\
&\to \mathrm{Conv2d}_{16} \to \mathrm{LeakyReLU}_{0.2} \to \mathrm{AvgPool}_2 \\
&\to \mathrm{Conv2d}_{32} \to \mathrm{LeakyReLU}_{0.2} \\
&\to \mathrm{Conv2d}_{32} \to \mathrm{LeakyReLU}_{0.2} \to \mathrm{AvgPool}_2 \\
&\to \mathrm{Conv2d}_{64} \to \mathrm{LeakyReLU}_{0.2} \\
&\to \mathrm{Conv2d}_{64} \to \mathrm{LeakyReLU}_{0.2} \to \mathrm{AvgPool}_2 \\
&\to \mathrm{Flatten} \to \mathrm{FC}_{128} \to \mathrm{ReLU} \\
&\to \mathrm{FC}_{d_Z} \to \ell^2 \text{ normalization}
\end{aligned}
$$

where $d_Z$ is the dimension of the latent space (either 11 for $S^{10}$ or 3 for $S^2$).

The architecture of the decoder is

$$
\begin{aligned}
z \in \mathbb{R}^{d_Z} &\to \mathrm{FC}_{128} \to \mathrm{FC}_{1024} \to \mathrm{ReLU} \\
&\to \mathrm{Reshape(64x4x4)} \to \mathrm{Upsample}_2 \to \mathrm{Conv}_{64} \to \mathrm{LeakyReLU}_{0.2} \\
&\to \mathrm{Conv}_{64} \to \mathrm{LeakyReLU}_{0.2} \\
&\to \mathrm{Upsample}_2 \to \mathrm{Conv}_{64} \to \mathrm{LeakyReLU}_{0.2} \\
&\to \mathrm{Conv}_{32} \to \mathrm{LeakyReLU}_{0.2} \\
&\to \mathrm{Upsample}_2 \to \mathrm{Conv}_{32} \to \mathrm{LeakyReLU}_{0.2} \\
&\to \mathrm{Conv}_1 \to \mathrm{Sigmoid}
\end{aligned}
$$

To compare the different autoencoders, we used as the reconstruction loss the binary cross entropy, $\lambda = 10$, Adam (Kingma & Ba, 2014) as optimizer with a learning rate of $10^{-3}$ and Pytorch's default momentum parameters for 800 epochs with batch of size $n = 500$. Moreover, when using SW type of distance, we approximated it with $L = 1000$ projections.

For the experiment on CIFAR10, we use the same architecture as Tolstikhin et al. (2018). More precisely, the architecture of the encoder is

$$
\begin{aligned}
x \in \mathbb{R}^{3\times32\times32} &\to \mathrm{Conv2d}_{128} \to \mathrm{BatchNorm} \to \mathrm{ReLU} \\
&\to \mathrm{Conv2d}_{256} \to \mathrm{BatchNorm} \to \mathrm{ReLU} \\
&\to \mathrm{Conv2d}_{512} \to \mathrm{BatchNorm} \to \mathrm{ReLU} \\
&\to \mathrm{Conv2d}_{1024} \to \mathrm{BatchNorm} \to \mathrm{ReLU} \\
&\to \mathrm{FC}_{d_z} \to \ell^2 \text{ normalization}
\end{aligned}
$$

where $d_z = 65$.

The architecture of the decoder is

$$z \in \mathbb{R}^{d_z} \rightarrow \mathrm{FC}_{4096} \rightarrow \mathrm{Reshape}(1024 \times 2 \times 2)$$
$$\rightarrow \mathrm{Conv2dT}_{512} \rightarrow \mathrm{BatchNorm} \rightarrow \mathrm{ReLU}$$
$$\rightarrow \mathrm{Conv2dT}_{256} \rightarrow \mathrm{BatchNorm} \rightarrow \mathrm{ReLU}$$
$$\rightarrow \mathrm{Conv2dT}_{128} \rightarrow \mathrm{BatchNorm} \rightarrow \mathrm{ReLU}$$
$$\rightarrow \mathrm{Conv2dT}_{3} \rightarrow \mathrm{Sigmoid}$$

We use here a batch size of $n = 128$, $\lambda = 0.1$, the binary cross entropy as reconstruction loss and Adam as optimizer with a learning rate of $10^{-3}$.

We report in Table 2 the FID obtained using 10000 samples and we report the mean over 5 trainings.

For SSW, we used the formulation using the uniform distribution (12). To compute SW, we used the POT library (Flamary et al., 2021). To compute the Sinkhorn divergence, we used the GeomLoss package (Feydy et al., 2019).

**Additional experiments.** We report on Figure 18 samples obtained with SSW for a uniform prior on $S^{10}$.

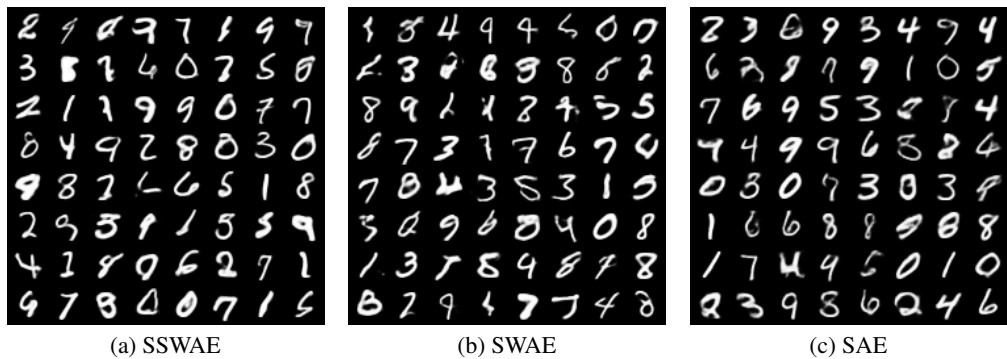

(a) SSWAE  (b) SWAE  (c) SAE

Figure 18: Samples generated with Sliced-Wasserstein Autoencoders with a uniform prior on $S^{10}$.

On Figure 19, we add the evolution over epochs of the Wasserstein distance between generated images and samples from the test set.

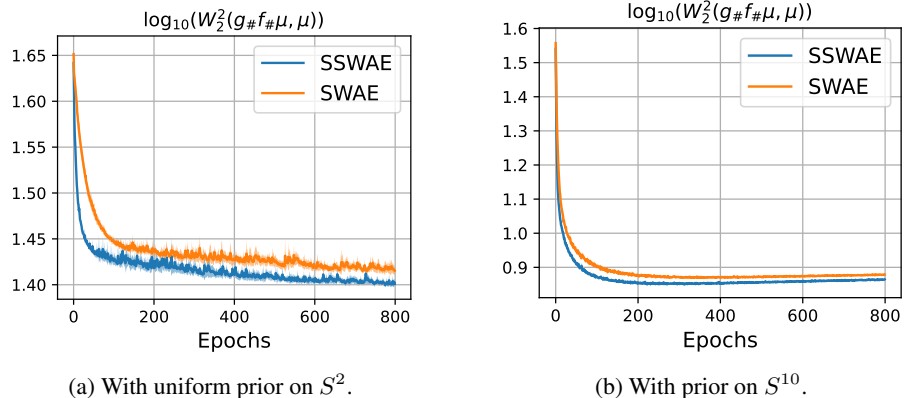

(a) With uniform prior on $S^2$.  (b) With prior on $S^{10}$.

Figure 19: Comparison of the evolution of the Wasserstein distance over epochs between SWAE and SSWAE on MNIST (averaged over 5 trainings).

## C.7 Self-supervised Learning

We conduct experiments using SSW to prevent collapsing representations in contrastive self-supervised learning (SSL) models. Such contrastive losses on the hypersphere have exhibited great representative capacity (Wu et al., 2018; Chen et al., 2020a; Caron et al., 2020) on unlabelled datasets by learning robust image representations invariantly to augmentations. As proposed in (Wang & Isola, 2020), the contrastive objective can be decomposed into an alignment loss which forces positive representations coming from the same

Table 4: Linear evaluation on CIFAR10. The features are taken either on the encoder output or directly on the sphere $S^2$.

| Method | Encoder output | $S^2$ |
|---|---|---|
| Supervised | 82.26 | 81.43 |
| Chen et al. (2020a) | 66.55 | 59.09 |
| Wang & Isola (2020) | 60.53 | 55.86 |
| SW-SSL, $\lambda = 1, L = 10$ | 62.65 | 57.77 |
| SW-SSL, $\lambda = 1, L = 3$ | 62.46 | 57.64 |
| SSW-SSL, $\lambda = 20, L = 10$ | 64.89 | 58.91 |
| SSW-SSL, $\lambda = 20, L = 3$ | 63.75 | 59.75 |

image to be similar and a uniformity loss which preserves maximal information of the feature distribution and hence avoids collapsing representations. Without the uniformity loss, the representations tend to converge towards a constant representation which yields the best alignment loss possible but also contains no information about original images. Wang & Isola (2020) propose to enforce uniformicity by leveraging the Gaussian potential kernel which is bound to the uniform distribution on the sphere. This formulation is also related to the denominator of the contrastive loss as specified in Chen et al. (2020a). We propose to replace the Gaussian kernel uniformity loss with SSW for which the complexity is more linear *w.r.t.* the number of batch samples. A simple choice of the alignment loss is to minimize the mean squared euclidean distance between pairs of different augmented versions of the same image. A self-supervised learning network is pre-trained using this alignment loss added with an uniformity term. Our overall self-supervised loss can be defined as:

$$\mathcal{L}_{\text{SSW-SSL}} = \underbrace{\frac{1}{n}\sum_{i=1}^{n}\|z_i^A - z_i^B\|_2^2}_{\text{Alignment loss}} + \frac{\lambda}{2}\big(\underbrace{SSW_2^2(z^A, \nu) + SSW_2^2(z^B, \nu)}_{\text{Uniformity loss}}\big), \tag{86}$$

where $z^A, z^B \in \mathbb{R}^{n \times d}$ are the representations from the network projected on the hypersphere of two augmented versions of the same images, $\nu = \text{Unif}(S^{d-1})$ is the uniform distribution on the hypersphere and $\lambda > 0$ is used to balance the two terms.

We pretrain a ResNet18 (He et al., 2016) model on the CIFAR10 (Krizhevsky, 2009) data with projections projected onto the sphere $S^2$. This feature dimension allow us to visualize the entire validation set of CIFAR10 and its distribution on the sphere. The visualization of the projections on $S^2$ are visible on Figure 20. We then evaluate the performance of each contrastive objective by fitting a linear classifier on top of the output of the layer before the projection on the sphere on the training dataset as is common for SSL methods. For comparison, we also report the results when the features are taken directly on the sphere. As a baseline, we also train a predictive supervised encoder by training jointly the linear classifier and the image encoder in a supervised manner using cross entropy.

We use a ResNet18 (He et al., 2016) encoder which outputs 1024 features that are then projected onto the sphere $S^2$ using a last fully connected layer followed by a $\ell^2$ normalization. We pretrain the model for 200 epochs using minibatch stochastic gradient descent (SGD) with a momentum of 0.9, a weight decay of 0.001 and an initial learning rate of 0.05. We use a batch size of 512 samples. The images are augmented using a standard set of random augmentations for SSL: random crops, horizontal flipping, color jittering and gray scale transformation as done in Wang & Isola (2020). For the trade-off parameter $\lambda$, we $\lambda = 20$ for SSW and $\lambda = 1$ for SW.

To evaluate the performance of representations, we use the common linear evaluation protocol where a linear classifier is fitted on top of the pre-trained representations and the best validation accuracy is reported. The linear classifiers are trained for 100 epochs using the Adam (Kingma & Ba, 2014) optimizer with a learning rate of 0.001 with a decay of 0.2 at epoch 60 and 80. We compare our methods with two other contrastive objectives, Chen et al. (2020a) with the normalized temperature-scaled cross-entropy (NT-Xent) loss and Wang & Isola (2020) which proposes to decompose the

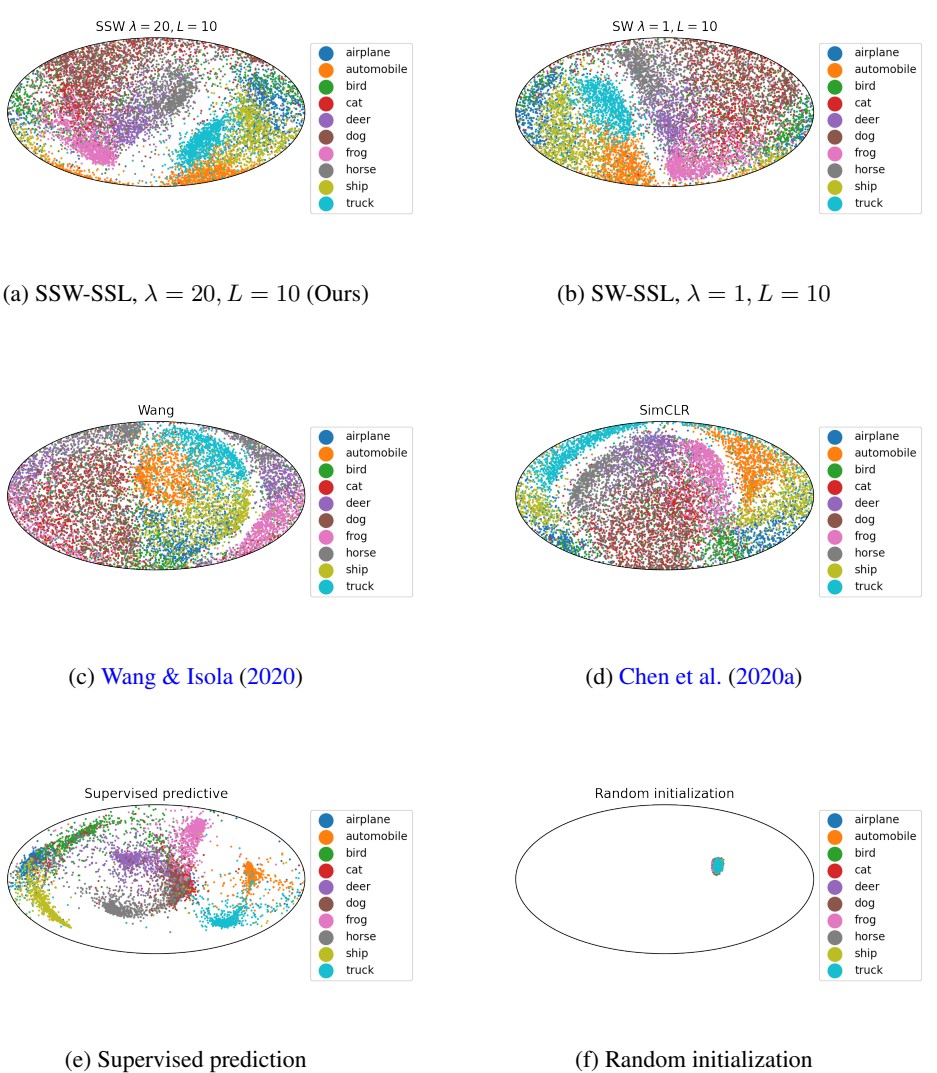

(a) SSW-SSL, $\lambda = 20$, $L = 10$ (Ours)

(b) SW-SSL, $\lambda = 1$, $L = 10$

(c) Wang & Isola (2020)

(d) Chen et al. (2020a)

(e) Supervised prediction

(f) Random initialization

Figure 20: The CIFAR10 validation set on $S^2$ after pre-training.

Table 5: Comparison of contrastive methods and their respective uniformity objective where $z^A, z^B \in \mathbb{R}^{n \times d}$ are representations from two augmented versions of the same set of images and $\nu = \mathrm{Unif}(S^{d-1})$ is the uniform distribution on the hypersphere.

| Method | $\mathcal{L}_{\mathrm{uniform}}(z^A) + \mathcal{L}_{\mathrm{uniform}}(z^B)$ | Complexity |
|---:|:---:|:---:|
| Chen et al. (2020a) | $\frac{1}{2n}\sum_{i=1}^{n}\log\sum_{j\neq i}\exp(\frac{\langle \hat{z}_i, \hat{z}_j\rangle}{\tau}), \hat{z}=\mathrm{cat}(z^A, z^B)$ | $O(n^2 d)$ |
| Wang & Isola (2020) | $\sum_{z\in\{z^A, z^B\}}\log\frac{2}{n(n-1)}\sum_{i>j}\exp(-t\|z_i - z_j\|_2^2)$ | $O(n^2 d)$ |
| SSW-SSL (Ours) | $\frac{1}{2}(SSW_2^2(z^A, \nu) + SSW_2^2(z^B, \nu))$ | $O(Ln(d + \log n))$ |

objective in two distinct terms $\mathcal{L}_{\mathrm{align}}$ and $\mathcal{L}_{\mathrm{uniform}}$. We recall the respective uniformity loss of each method in Table 5. As one can see in Table 4, our method achieves here comparable performances to two state-of-the-art approaches, yet slightly under-performing compared to (Chen et al., 2020a). We suspect that a finer validation of the balancing parameter $\lambda$ is needed. Especially since the representations on Figure 20a are not completely uniformly distributed around the sphere after pre-training compared to other contrastive methods. Nevertheless, these preliminary results show that SSW-SSL is a promising contrastive learning approach without explicit distances between negative samples, especially compared to SW on the sphere. To this end, further works should be devoted to finding a good balance between the alignment and uniformity objectives.

