# OpenReview forum: "Spherical Sliced-Wasserstein"
_ICLR.cc/2023/Conference — ICLR 2023 poster_

### Official Review · Reviewer_NCpB · 2022-10-18

**Confidence:** 4
**Correctness:** 3
**Technical Novelty And Significance:** 2
**Empirical Novelty And Significance:** 3
**Recommendation:** 6

**Clarity, Quality, Novelty And Reproducibility:**

# Clarity

As said above, the paper is very clear and well-written.

# Quality

I think this is a competent paper, the proposed approach is robust and motivated, and experiments are convincing overall.

# Novelty

As said above, the proposed approach is novel to the best of my knowledge, though it is naturally adapted from the usual SW distance. I overall think that the novelty is somewhat limited. See **Weaknesses** above.

# Reproducibility

The clarity of the paper makes the work reasonably easy to reproduce as far as I can tell.
Some experimental details in the **Runtime Comparison** paragraph are lacking though. Saying that SSW is faster (or not) than Sinkhorn is not very inisghtful given that these algorithms compute different things. Sinkhorn approximates the true Wasserstein distance while "Monte-Carlo SSW" approximate the true SSW which, as far as I can tell, is not directly related to the true Wasserstein distance between the two measures.

**Details Of Ethics Concerns:**

No concern specific to this work.

**Strength And Weaknesses:**

# Strengths :
- The paper is overall very well written, with appropriate level of mathematical rigor (though this may be a matter of taste).
- The construction is very natural yet (perhaps surprisingly) new to the best of my knowledge.
- While I did not proofread the appendices in full details, proofs seems well-written and I did not identify a major flaw at first glance.
- [minor but still] The figures are of high-quality.

# Weaknesses :
- (1) To some extend, one may argue that the contributions of the work are somewhat limited, in sense that the construction follows the exact step of the usual SW distance and, up to few technical modification brought to the Radon transform, there is no specific "breakthrough" in this paper.
- (2) The setting is restricted to the spherical case. While this yield elegant formulas and possible implementation overall, it is also somewhat disappointing in sense that the proposed method may be generalized quite well (typically intersecting [insert assumptions here] hypersurfaces with $V_{d,2}$). From a theoretical standpoint, that may lead to a more instructive construction.
- (3) A reference and discussion to the recent preprint _Intrinsic sliced wasserstein distances for comparing collections of probability distributions on manifolds and graphs_, Rustamov et al., may be relevant. In this work as well, the authors build a SW-like distance on manifolds. They follow a fairly different approach though (relying on the eigenfunctions of the Laplace-Beltrami operator on the manifold). It is not clear to me how the two methods compare. While the approach proposed in the current paper seems (to me) more natural and easier to implement, the one of Rustamov et al., has the benefits of being more general and quite original.

**Summary Of The Paper:**

Following the well-established Sliced-Wasserstein (SW) used as a Wasserstein-based metric to compare measures supported on the Euclidean space $\mathbb{R}^d$, this paper introduces a natural Spherical-Sliced-Wasserstein (SSW) distance to compare measures supported on an hypersphere.

The construction technique follows the SW-one, namely :
- First, observe that comparing measures supported on the circle $S^1$ can be done fairly efficiently (akin to the comparison of measures supported on $\mathbb{R}$).
- In dimension $d$, intersect the hypersphere $S^{d-1}$ with 2d-planes (parametrized by the Stiefel manifold $V_{d,2}$) to get a (great) circle, project measures of interest $\mu,\nu$ on these circles, compute (efficiently) the Wasserstein distance between those measures. Eventually, integrate over all the circles.

The authors show that this construction defines a pseudo-metric on $P(S^{d-1})$ (probability measures supported on the hypersphere) ; provide implementation details and showcase their approach in a variety of experiments.

**Summary Of The Review:**

I think this is a well-written paper that present a natural, well-motivated and easy-to-understand approach.

On the downsides, I would say that the paper does not bring any brand new idea from the theoretical side that would shed a new light on "numerical OT on manifolds" ; and while the experiments are interesting, I cannot assess the potential impact for numerical OT practitioners (though it clearly is non-zero).

---

> ### Author Response · Authors · 2022-11-10
> **Response to Reviewer NCpB**
>
> We thank the reviewer for their comments. We address their concern below.
>
> **Contributions of the work limited**: We agree that the construction is relatively natural as it uses intrinsic counterparts on the sphere of objects used on Euclidean space for SW. However, we believe that it is still a novel and interesting object to study. Moreover, it raises a number of questions which are more complicated than on Euclidean space.
>
> We also expect that this family of methods can be transposed to other types of manifolds, such as hyperbolic spaces, and we hope to pave the way for such future works.
>
>
> **The setting is restricted to spherical case**: We agree that a more general theory would be of much interest and we will surely consider it in future works. But we still would like to stress that the spherical case already raises theoretical open questions which are not straightforward to deal with.
>
>
> **Reference [1]**: We thank you for this reference that we were not aware of and which is clearly relevant to our work. We discussed it in the revised version of the paper.
>
> As underlined by the reviewer, the methods are indeed very different. In our work, we use counterparts of objects used in the Euclidean space for SW to define SSW, while in [1], they use eigenvalues and eigenfunctions of the Laplace-Beltrami operator. The eigenfunctions are used to project measures on the real line, while the eigenvalues weight the Wasserstein distance corresponding to each eigenfunction. Altough it is a very interesting proposal, this construction is not really in the same spirit of SW, but more in the spirit of max-K SW where we project only over a delimited number of directions. In particular, using their method on Euclidean space would not correspond to the usual SW method. Hence, their method actually defines a new SW distance on $\mathcal{P}(\mathbb{R}^d)$.
>
> It seems that [1] is not yet published and that no official code is available. We tried a quick implementation using the code of [2] to compute the spherical harmonics (which are the eigenfunctions of the Laplace-Beltrami operator on the sphere). We noted that it requires more hyperparameters to tune. Indeed, we need to choose the number of eigenfunctions, but also the kernel applied on eigenvalues to weight the Wasserstein distances as well as kernel hyperparameters. Given the limited time for the rebuttal, we could not tune the hyperparameters to make it work well on e.g. the density estimation experiment. We report the results obtained using the first 20 eigenfunctions, and the heat kernel with $t=0.5$ on the following table:
>
> |  | Earthquake | Flood | Fire |
> |--------|----------------|-------|---|
> | SSW | $0.84\pm 0.07$ | $1.26\pm 0.05$ | $0.23\pm 0.18$ |
> | ISW [1] | $2.70 \pm 0.54$ | $2.51\pm 0.16$ | $2.05\pm 0.46$ |
>
> We also would like to note that their method might be complicated to compute in dimension $d\ge 10$ as computing the spherical harmonics can become unstable (Appendix A of [2]).
>
> **SSW faster than Sinkhorn is not very insightful**: We kindly disagree on this statement. Indeed, the entropic regularization is often used in practice as a proxy of the Wasserstein distance for different learning tasks, as it is differentiable and has a better complexity w.r.t. to the number of samples. From that point of view, SSW is another alternative which can be used in place of Sinkhorn, and which is computationally more efficient.
>
> We agree that SSW might not be related to the true geodesic Wasserstein distance, but as it is an OT based discrepancy, we believe that the comparison can be done.
>
>
> [1] Rustamov, R. M., & Majumdar, S. (2020). Intrinsic sliced wasserstein distances for comparing collections of probability distributions on manifolds and graphs. arXiv preprint arXiv:2010.15285.
>
> [2] Dutordoir, Vincent, Nicolas Durrande, and James Hensman. "Sparse Gaussian processes with spherical harmonic features." International Conference on Machine Learning. PMLR, 2020.

---

> > ### Comment · Reviewer_NCpB · 2022-11-15
> > **Thanks for addressing my review**
> >
> > > (...) it is still a novel and interesting object to study. Moreover, it raises a number of questions which are more complicated than on Euclidean space.
> >
> > Agreed. This model gives some further motivation to work beyond the Euclidean setting.
> >
> > > But we still would like to stress that the spherical case already raises theoretical open questions which are not straightforward to deal with.
> >
> > Can you elaborate on this? From my understanding, the importance of the spherical setting is that you have access to a (almost) closed-form for the corresponding 1D transport cost (and also geodesic projection and so on). From the theoretical viewpoint, what properties of the sphere are important to define the SSW? Wouldn't the construction adapts at least to any manifold that are diffeomorphic to the sphere?
> >
> > > their method on Euclidean space would not correspond to the usual SW method.
> >
> > Agreed. Thank you for detailing the comparison.
> >
> > > SSW vs Sinkhorn
> >
> > I completely agree with the fact that Sinkhorn is widely used as a way to approximate Wasserstein distances. My claim, as you said, is that (S)SW is **not** approximating the Wasserstein distance. Hence, the comparison is hardly fair: you cannot make a proper statement/experiment saying "Method 1 provides an $\epsilon$ approximation of quantity $Q$ in $t_1$ time while method 2 provides an $\epsilon$ approximation of $Q$ in $t_2$ time".
> >
> > For instance, if the goal of the downstream task is to estimate the exact Wasserstein distance, then Sinkhorn would be better than (S)SW in most regimes, no matter the running time.
> > Maybe a fair way to empirically compare both approaches is to consider a two-sample-test task (or another task where Wasserstein/Sinkhorn is known to be useful), and to show that SSW can achieve similar performances in a much lower time (or much better performances in the same time).
> >
> > I am not saying that this should necessarily be done in the current work (though it would be an interesting addition), but just that the takeaway from Figure 2 remains unclear to me.
> >
> > _Note :_ If my understanding is correct, one rigorous way to support (S)SW over Sinkhorn is that the former scales roughly in $O(n \log n)$ while the latter scales roughly in $O(n^2)$, with respect to the sample size $n$.

---

> > > ### Author Response · Authors · 2022-11-16
> > > **Response to Reviewer NCpB**
> > >
> > > Thank you for response.
> > >
> > > **Theoretical questions raised in the spherical case**: You are right, the importance of the spherical setting is the access to closed-forms for the projections and 1D transport. We rather meant that there are still some theoretical open questions on SSW such as being or not a distance, or having or not topological properties analogous to the Wasserstein distance with geodesic distance on the sphere.
> > >
> > > From a theoretical viewpoint, a guess is that we can always use the geodesic projection and integrate over geodesics to define a sliced version. However, in practice, we need to derive for each case considered the corresponding geodesics and projections, as well as the right Wasserstein distance in 1D. This might not be straightforward. Indeed, for example for a torus, there are several types of geodesics (periodic or infinite), and some of them cannot be obtained by just intersecting with $\mathbb{V}_{d,2}$. Another example is the Stiefel manifold, which is very related to the sphere, but for which we actually do not have closed-forms for its geodesics [1].
> > >
> > > Finally, for now, we do not have theoretical results relying on specifically using the geodesic projection, as our theoretical results are rather general to any slicing method. Hence, we believe that focusing on the sphere is already challenging enough, and studying a more generalized formulation might complicate things for now. Nevertheless, we agree that a more general construction would be of much interest, and we will definitely look into it in future works.
> > >
> > > **Wouldn't the construction adapt at least to any manifold that are diffeomorphic to the sphere?**
> > > If we understand the reviewer's point well, the idea would be to either characterize or to construct an invertible and differentiable mapping from a given manifold to the sphere, and then use the SSW directly on the image measure through this mapping on the Sphere. This is very interesting, and deserves more thinking. Notably, as this diffeomorphism would generally not be an isometry, the corresponding sliced distance would be different from the true geodesic one in the original manifold. As a simple example, one can consider a conformal map between a plane and the sphere, such as a stereographic projection, which is a diffeomorphism from a hyperplane to the whole sphere except a pole. However, the constructions of sliced distances in these two spaces are differents and not equivalents as the stereographic projection is not an isometry. Let us finally note that the converse is also possible (mapping from the Sphere to a hyperplane, then considering the sliced Wasserstein distance in the hyperplane). In our experiment from Table 1 however we noted that this strategy did not give satisfying results. We believe that in some sense the preservation of the geometry (distances) is of prime importance when designing sliced wasserstein distances onto manifolds.
> > >
> > > **SSW vs Sinkhorn**: Entropy regularized OT (EOT) and the corresponding Sinkhorn algorithm are usually used by practioners in many problems in place of the regular OT problem (even if the two problems are different and do not yield the same solution, apart when the regularization tends to zero). EOT per se is not a distance (it can be negative due to the entropic term), and generally one would prefer sharp Sinkhorn [2] or Sinkhorn divergence [3] as a surrogate if the distance property is needed. Yet, Sinkhorn (as SW) is definitely a good choice when used as **a loss function** which is differentiable. The goal of Figure 2 is **not** to assess if Sinkhorn or SSW give meaningful approximations of the true geodesic Wasserstein distance, but rather to compare runtimes for a given number of samples, and hence the complexity ($O(n\log n)$ for SSW vs $O(n^2)$ for Sinkhorn). We also discuss the fact that SSW scales better in term of memory (since we do not store the matrix of distances, nor the full coupling matrix). In our view, Figure 2 is important to highlight the main advantage wrt. the exact Wasserstein distance (and EOT) which lies on the computational efficiency. We also thank you for the suggestion of the statistical test which might be a good way of comparing the real performances of the different metrics.
> > >
> > >
> > > [1] Chakraborty, Rudrasis, and Baba C. Vemuri. "Statistics on the Stiefel manifold: theory and applications." The Annals of Statistics 47.1 (2019): 415-438.
> > >
> > > [2] G Luise, A Rudi, M Pontil, C Ciliberto. "Differential properties of sinkhorn approximation for learning with wasserstein distance". Advances in Neural Information Processing Systems, 2018
> > >
> > > [3] Jean Feydy, Thibault Séjourné, François-Xavier Vialard, Shun-ichi Amari, Alain Trouvé, Gabriel Peyré "Interpolating between optimal transport and mmd using sinkhorn divergences".The 22nd International Conference on Artificial Intelligence and Statistics

---

### Official Review · Reviewer_sqi7 · 2022-10-23

**Confidence:** 5
**Correctness:** 4
**Technical Novelty And Significance:** 3
**Empirical Novelty And Significance:** 3
**Recommendation:** 8

**Clarity, Quality, Novelty And Reproducibility:**

* Clarity is one of the strengths of this paper.

* While the transport-based distances on circles have been studied before, this paper introduces a novel spherical slicing that enables the definition of SSW. Hence, the proposed approach has sufficient novelty.

* The algorithmic simplicity of the approach allows for a transparent implementation of the proposed pseudometric. In addition, the code was provided by the authors, and I found the results easily reproducible.

**Details Of Ethics Concerns:**

No ethics concerns.

**Strength And Weaknesses:**

### Strengths

* The paper addresses the exciting problem of comparing distributions defined on a hypersphere, which appears in many applications.
* The paper introduces a simple and computationally efficient pseudometric for comparing spherical distributions that is theoretically sound.
* The paper is very well written, and it is easy to follow.
* The authors show the application of the proposed SSW on a wide variety of applications. What I particularly appreciate is that while some of the experiments are not large-scale, they remain challenging to solve by existing methods.

### Weaknesses

I don't see any significant weaknesses in this paper. Below are not necessarily weaknesses but points that could improve the paper.

* It would have been great if the authors could show that the proposed Radon transform is invertible, and hence SSW is indeed a metric as opposed to a pseudometric.
* I am surprised that SW distance works so well compared to SSW for spherical distributions. I was expecting a more significant performance gap.



**Summary Of The Paper:**

The paper introduces Spherical Sliced-Wasserstein (SSW) as a "pseudometric" to measure the dissimilarity between distributions defined on a hypersphere. At its core, the paper utilizes the prior work proposed by Rabin et al. 2011a that studies transport-based distances on circles and proposes efficient numerical schemes for calculating these distances. The authors then propose to slice distributions defined on a hypersphere by projecting them onto the great circles leading to spherical slices. Lastly, following the vanilla-sliced Wasserstein framework, the Spherical Sliced-Wasserstein is defined as the expected Wasserstein distance between the spherical slices of two distributions. According to Rabin et al. 2011a, the 1-Wasserstein distance between two distributions defined on a circle has a closed-form solution and can be obtained efficiently.  In addition, the authors show that the 2-Wasserstein distance between a distribution defined on a circle and the uniform distribution also has a closed-form and computationally efficient solution (Proposition 1).

The paper provides extensive experiments (both in the main paper and in the appendix) showing that the proposed SSW "pseudometric" is effective in various applications, from density estimation to gradient flow and learning auto-encoders. The appendix also provides additional results on, for instance, self-supervised learning with spherical latent representations. The results indicate the robustness and effectiveness the proposed dissimilarity measure.

**Summary Of The Review:**

Overall, I think this is a fantastic paper. Here are my primary considerations in evaluating the work:

* It is well-motivated, clear, and a pleasant read
* It is algorithmically simple while being theoretically sound
* It showcases the application of the proposed approach to a wide range of problems.

While the experiments might be far from being large-scale, they remain relevant and challenging.

---

> ### Author Response · Authors · 2022-11-10
> **Response to Reviewer sqi7**
>
> We thank the reviewer for their enthusiastic comments. We answer the few concerns raised below.
>
> **Show that the Radon transform is invertible:** We agree that it would be great to show that the spherical Radon transform related to SSW is injective. We note that it is a difficult problem, only tackled by few specialists, and hence we did not succeed at showing this yet. See also our response to Reviewer 7Kyx.
>
> **Gap between SW and SSW:** The fact that SW performs actually fairly well on spherical data is very surprising indeed. However, working on the ambient space, SW is well defined on spherical distributions, and is still a distance between such distributions.

---

> > ### Comment · Reviewer_sqi7 · 2022-11-15
> > **Response to authors comments**
> >
> > I thank the authors for their response. I read my fellow reviewers' comments and the authors' responses.
> >
> > I still think this is a good paper, and I vote for its acceptance.

---

### Official Review · Reviewer_7Kyx · 2022-10-24

**Confidence:** 3
**Correctness:** 4
**Technical Novelty And Significance:** 3
**Empirical Novelty And Significance:** 3
**Recommendation:** 6

**Clarity, Quality, Novelty And Reproducibility:**

Overall, the paper is well written and easy to follow up. It thoroughly  reviews the background and main works on Sliced Wasserstein and Wasserstein on circle to introduce optimal transport on sphere and its sliced version computation. The proposed contributions follow the classical procedure of deriving efficient sliced optimal transport computation. However the SSW departs from existing works by addressing the challenging Wasserstein distance between distribution defined on sphere, paving the way to the derivation of computationally efficient algorithm for OT on manifold. As such the paper is of interest and the contributions are somehow significative.

**Strength And Weaknesses:**

**Strengths**
- The rationale behind the proposed Spherical Sliced-Wasserstein (SSW) method is justified.
-  The approach aims to design an appropriate geodesic projection on great circles of distributions defined over sphere. That projection acts as the projection on straight line used in Sliced Wasserstein (SW). Overall the projection operators are formulated as matrices over Stiefel manifold. Using known results on Wasserstein distance on circle, a sliced version of Wasserstein distance on sphere is derived.
- As a contribution, the paper proposes a closed-form projection of the training samples from the sphere to the great circle using matrix on Stiefel manifold.
- Another interesting contribution is the proposal of the spherical Radon transform and related dual operator that allow to rephrase SSW as a function of the new Radon transform.  The pseudo-distance property of SSW is established in Proposition 4.
- Intensive empirical evaluations are conducted. They show that SSW exhibit a favorable runtime in a large scale setting. Used as a loss function, SSW allows for density estimation of simulated and real datasets with advantageous empirical results. Also SSW proves effective in generative modeling applied to classical Mnist, Fashion Mnist and Cifar datasets.

**Weaknesses**
- The plain metric property of SSW is not guaranteed.
- Topological properties of SSW compared to plain Wasserstein distance on sphere are not discussed. Can one theoretically or empirically relate SSW to its plain counterpart?
- For readability purpose, it would be interesting to formally introduce the notion of great circle in the paper.

*Minor comments*

- In Figure 4 illustration, which dataset is shown?
- On page 8, the sentence "We see on this figure that the normalizing flows fitted are able to recover the principles modes of the data" is not clear. Where exactly on Figure 5 the recovery of the modes are highlighted?
- The paper should better introduce (at least in the supplementary material) some preliminary notions of Wasserstein distance on the circle. Especially, the parametrization by the interval [0, 1[ on the real line should be better explained.
- Some notations or abbreviations have to be clarified or formally defined within the main text. For instance, in equation (6), $C(\mathbb{R} \times S^{d-1})$ is not stated. Page 4, what $\mathcal{P}_{p, ac}$ stands for? Page 9, NF is for normalizing flows?

**Summary Of The Paper:**

The paper proposes a slice-based approach to efficiently compute the Wasserstein distance between two distributions $\nu$ and $\mu$ defined on the sphere. The method termed SSW (Spherical  Sliced Wasserstein) projects the distributions on great spheres and then, hinges on the efficient computation of Wasserstein distance on the circle to propose the sliced-variant of Wasserstein distance on the sphere. Along with SSW, the related spherical Radon Transform and its dual operator are formulated. Finally it is established that the proposed SSW defines a pseudo-distance. A numerical algorithm that summarizes the design of the random projection matrices, the computation of the samples' coordinate on the circle as well as the Wasserstein distance on the circle. Empirical evaluations on simulation dataset or on learning latent representations via auto-encoders highlight the potential of the proposed method.

**Summary Of The Review:**

The paper is well written and proposes a new sliced optimal transport on a sphere. It also proposes the related spherical Radon Transform. The new discrepancy measure of distributions over sphere is supported by its pseudo-metric property and empirical experiments showing its effectiveness. Topological properties of the proposed approach and some few details on the empirical  evaluations remain to be clarified.

---

> ### Author Response · Authors · 2022-11-10
> **Response to Reviewer 7Kyx**
>
> We thank the reviewer for their comments. We address their concerns below.
>
> **The plain metric property of SSW not guaranteed:** We agree that it is somewhat underwhelming to not know whether or not it is a metric. In our case, the only missing property is the identity of indiscernibles, which is usually related with the Radon transform. Hence, we started to make connections with a related spherical Radon transform which injectivity would help to show that it is a metric. However, it is an intricate field tackled only by few specialists, and we were not able to solve this problem for the moment. We hope to spark interest on this problem.
>
> Let us note that in practice, we never observed such a case where $SSW(\mu,\nu)=0$ with $\mu \neq \nu$. We might conjecture that if there exists such cases, they can be only pathological.
>
> **Topological properties not discussed:** We agree that it is an important property of SW on Euclidean spaces, which we would like to have for SSW.
>
> For SW in the Euclidean case, it is proved in Theorem 5.1.5 of [1] that for $\mu$ and $\nu$ compactly supported,
> \begin{equation}
>     SW_p^p(\mu,\nu) \le c_{d,p} W_p^p(\mu,\nu) \le C_{d,p} SW_p(\mu,\nu)^{\frac{1}{d+1}},
> \end{equation}
> which means that $SW_p$ and $W_p$ are topologically equivalent. The first inequality is obtained by using that the linear projection is 1-Lipschitz. This does not seem to be the case for the geodesic projection on great circles. The second inequality is obtained by exploiting the relation between SW and the Fourier transform. But the geodesic projection used here does not seem to share many links with the spherical harmonic Fourier transform.
>
> A second proof of the topological equivalence for general measures is available in Theorem 1 of [2]. But this proof heavily relies on the characteristic function and on Levy's characterization theorem. The characteristic function is basically the Fourier transform of the probability measures. Hence, we have the same issues than for the proof of [1].
>
> For these different reasons, showing the topological equivalence between SSW and W with the geodesic distance on the sphere seems out of reach for now.
>
>
> **Introduce great circles:** We introduced the notion of great circle in the revision of the paper.
>
> **Figure 4:** The dataset used for Figure 4 is MNIST. This was added in the revised version of the paper.
>
> **Where exactly on Figure 5 the recovery of the modes are highlighted?**: The true densities of the datasets used in Figure 5 are not known. However, we plotted the density of the test data, and we see in red the regions where the learned model put the most mass. It actually corresponds to regions where most data points lie but we agree it might not be very clear, and we clarified this in the revised version.
>
>
> **Wasserstein on the circle:** We introduce the Wasserstein distance on the circle in Section 3.1 by shortly reviewing the relevant references. More details such as more explanations on the parameterization of $S^1$ by $[0,1[$ can be found in these papers (e.g. in [3,4]).
>
> **Abbreviations:** Thank you for spotting this, we corrected these abbreviations in the revised version of the paper.
>
>
> [1] Bonnotte, Nicolas. Unidimensional and evolution methods for optimal transportation. Diss. Paris 11, 2013.
>
> [2] Nadjahi, Kimia, et al. "Asymptotic guarantees for learning generative models with the sliced-wasserstein distance." Advances in Neural Information Processing Systems 32 (2019).
>
> [3] Rabin, Julien, Julie Delon, and Yann Gousseau. "Transportation distances on the circle." Journal of Mathematical Imaging and Vision 41.1 (2011): 147-167.
>
> [4] Hundrieser, Shayan, Marcel Klatt, and Axel Munk. "The statistics of circular optimal transport." Directional Statistics for Innovative Applications. Springer, Singapore, 2022. 57-82.

---

### Official Review · Reviewer_dqQ6 · 2022-10-24

**Confidence:** 4
**Correctness:** 4
**Technical Novelty And Significance:** 3
**Empirical Novelty And Significance:** 2
**Recommendation:** 6

**Clarity, Quality, Novelty And Reproducibility:**

The paper is well-written, the proposed SSW is clearly explained, this work is the first step towards slice-based Wasserstein distances on spherical manifolds. The experiment results can be reproduced.

**Strength And Weaknesses:**

Strength:

The SSW is the first step towards defining slice-based Wasserstein distances for measures defined on manifolds. Closed-form solutions to projections onto great circles are derived in the paper and subsequently the spherical Radon transform is also defined analogously to the standard Radon transform.

Weakness:

There are a few things that can be improved in the paper.
(1) The projection efficiency is not mentioned in the paper, but this is something important to slice-based Wasserstein distances. Can we quantify how many projections are needed to gain a good approximation of the distance?
(2) What is the computational cost of SSW compared to the standard SW? Runtime comparison shown in Figure 2 does not include standard SW. From the experiments, the standard SW also produced competitive results.
(3) It is not explained clearly in the paper why the SSW has better performance than the standard SW. The overall argument is that the data manifold is taken into consideration in SSW, but detailed discussions and analysis to support this claim with visualisations, experiments, and theoretical analysis will make it a stronger paper.


**Summary Of The Paper:**

A novel variant of slice-based Wasserstein distance is proposed in the paper, namely the spherical sliced Wasserstein (SSW) distance. The SSW is particularly suited to evaluating the discrepancy between probability measures defined on spherical manifolds. By projecting measures defined on hyperspheres onto great circles with the proposed spherical Radon transform, the SSW is defined as the expectation of Wasserstein distances between obtained measures on great circles w.r.t to the uniform distribution over the Stiefel manifold. The proposed SSW has lower computational costs compared to standard Wasserstein distances on $R^d$, since Wasserstein distances on circles admit closed-form solutions. Experiment results show that the SSW indeed can better capture geometric information from data than the original SW when the data of interest lies on spherical manifolds.

**Summary Of The Review:**

This paper introduced an interesting variant of SW distances which accommodates well with measures defined on spherical manifolds. The method is novel, and experiments show its effectiveness on several tasks.

---

> ### Author Response · Authors · 2022-11-10
> **Response to Reviewer dqQ6**
>
> We thank the reviewer for his/her comments. We address his/her questions and concerns below.
>
> **Projection efficiency**: We agree  that the projection efficiency is important to quantify how the number of Monte-Carlo projection impacts the quality of the estimate. By transposing the derivations of Theorem 6 in [1], we are able to show that, similarly to the classical SW distance, the squared projection complexity is bounded by the variance multiplied by $L^{-1}$. More precisely, we find that:
> \begin{equation}
>     \mathbb{E}_U\left[|\widehat{SSW}_\{p,L\}^p(\mu,\nu)-SSW_p^p(\mu,\nu)|\right]^2 \le \frac{1}{L}\mathrm{Var}_U\left(W_p^p(P^U_\\#\mu,P^U_\\#\nu)\right),
> \end{equation}
> where $\widehat{SSW}_\{p,L\}^p(\mu,\nu) = \frac{1}{L} \sum_\{i=1\}^L W_p^p(P^{U_i}_\\#\mu, P^{U_i}_\\#\nu)$ is the empirical approximation of SSW with $(U_i)_i\sim \sigma$ iid samples. We added this result in the appendix of the revised version.
>
> **Computational cost of SW:** As stated in Section 2.2, the complexity of SW is $O(Ln(d+ \log n))$. Hence, the complexity is the same as $SSW_2$ with the uniform distribution, and slightly lower compared to $SSW_1$ and $SSW_2$ with the binary search algorithm, as the computation of the Wasserstein distance in one dimension on the circle is slightly more complicated. We updated Figure 2 by adding the runtime of SW.
>
> Notably, as the geodesic projection on circles induces more operations compared to the linear projection of SW, we observe that for few samples, SW is faster. But when the cost of the sorting operation becomes higher than the cost of the projection, $SSW_2$ with the uniform distribution is faster to compute than SW. Moreover, up to a constant, the behavior of SW and SSW is the same.
>
> **Why SSW has better performance of SW:** This is a difficult question to answer. The main argument is that, as SSW is intrinsic to the sphere, it better takes into account the geometry of the manifold. However,  we still see that the gap between the two methods is fairly small. This may be explained as the sphere is a compact manifold, and hence as SW is still well defined on this manifold, it still works. We can expect that this will not be the case when considering other types of manifold, such as hyperbolic spaces which we will consider in future works.
>
> [1] Nadjahi, K., Durmus, A., Chizat, L., Kolouri, S., Shahrampour, S., & Simsekli, U. (2020). Statistical and topological properties of sliced probability divergences. Advances in Neural Information Processing Systems, 33, 20802-20812.

---

### Author Response · Authors · 2022-11-10
**General Response**

We thank the reviewers for their positive comments on our work. We added a revised version of the paper where we highlighted in red the changes.

---

### Decision · Program_Chairs · 2023-01-20

**Decision:**

Accept: poster

**Justification For Why Not Higher Score:**

The sliced Wasserstein is a popular topic in Wasserstein distance. The work presented in this paper is its variant. The idea is overall interesting, however, the number of potential researchers who would be interested in this work is slightly limited. Thus, I recommend it with Accept (poster).

**Justification For Why Not Lower Score:**

N/A

**Metareview: Summary, Strengths And Weaknesses:**

In this paper, the authors propose the Spherical Sliced-Wasserstein (SSW), which is a  "pseudometric" to measure the dissimilarity between distributions defined on a hypersphere. The idea of the sliced Wasserstein distance on a sphere is a new and novel. Moverover, the reviewers overall agree to accept the paper. So, I also vote for acceptance.

In the final version, we highly encourage authors update the paper based on the reviewer's comments.

**Note From Pc:**

if the above contains the word "oral" or "spotlight" please see: "oral" presentation means -> notable-top-5% and "spotlight" means -> notable-top-25%. As stated in our emails, we are disassociating presentation type from AC recommendations

**Summary Of Ac-Reviewer Meeting:**

N/A